# Adjusted Count Quantification Learning on Graphs

**Clemens Damke**
LMU Munich, MCML
clemens.damke@ifi.lmu.de

**Eyke Hüllermeier**
LMU Munich, MCML, DFKI
eyke@lmu.de

## Abstract

*Quantification learning* is the task of predicting the label distribution of a set of instances. We study this problem in the context of graph-structured data, where the instances are vertices. Previously, this problem has only been addressed via node clustering methods. In this paper, we extend the popular *Adjusted Classify & Count* (ACC) method to graphs. We show that the prior probability shift assumption upon which ACC relies is often not applicable to graph quantification problems. To address this issue, we propose *structural importance sampling* (SIS), the first graph quantification method that is applicable under (structural) covariate shift. Additionally, we propose *Neighborhood-aware ACC*, which improves quantification in the presence of non-homophilic edges. We show the effectiveness of our techniques on multiple graph quantification tasks.

## 1   Introduction

We consider the task of *quantification learning* (QL) on graph-structured data. This term was first coined by Forman [8, 9, 10] and is used to describe the task of estimating label prevalences via supervised learning. A QL method receives a set of training instances with known labels, which is used to train a quantifier. The quantifier is then used to predict the label distribution of a set of test instances. Unlike standard instance-wise classification, QL does not concern itself with predicting an accurate label for each test instance but rather with predicting the overall prevalence of each label across all instances. QL can thus be seen as a dataset-level prediction task, where a single prediction is made for a population of instances.

Quantification problems naturally arise in polling and surveying, where the goal is to estimate the proportion of a population that has a certain property or holds a certain opinion. Examples include estimating the proportion of voters who support a certain political party or the proportion of customers who are satisfied with a product. Similarly, QL can be applied to epidemiology or ecological modelling to estimate the prevalence of diseases or species in a given population. We refer to Esuli et al. [5] for a comprehensive overview of the applications of quantification.

Typically, QL is studied in the context of tabular data, where each instance $x \in \mathcal{X} = \mathbb{R}^d$ is represented by a feature vector. In this setting, instances are assumed to be independent, i.e., the label distribution $P(Y \mid X = x)$ is fully determined by the instance $x$. However, in many real-world applications, this independence assumption does not hold. Consider the example of estimating the proportion of voters supporting a certain party. Assume we have access to a social network where each node represents a voter and each edge represents a social connection. In this case, the label distribution of a voter, i.e., their political preferences, may depend not only on their own features but also on the features of their social connections. Incorporating this relational information into the quantification process can lead to more accurate estimates.

Generally speaking, QL methods can be divided into two categories: aggregative and non-aggregative. Aggregative quantifiers rely upon an instance-wise label estimator, i.e., a regular classifier; the instance-level label estimates are then aggregated to obtain dataset-level label prevalence

39th Conference on Neural Information Processing Systems (NeurIPS 2025).

estimates. Non-aggregative quantifiers, on the other hand, directly estimate dataset-level label prevalences without first predicting labels for each instance. In this paper, we focus on aggregative quantification methods, which are more common and have been studied more extensively. An intuitively plausible aggregative method is to simply estimate the prevalence of a label as the fraction of test instances that are predicted to belong to that label by the classifier. This method is known as *Classify & Count* (CC) and, given a perfect classifier, it will yield perfect quantification results. However, in practice, classifiers are not perfect, and even a good but not perfect classifier can lead to poor quantification results. Conversely, even a bad classifier can yield good quantification results. The reason for this disconnect is that the optimization goals of classification and quantification are misaligned. More specifically, while a good binary classifier should minimize the total number of misclassifications, i.e., $(FP + FN)$, a good binary quantifier should minimize $|FP - FN|$. If $FP = FN$, even a classifier with a high misclassification rate will yield perfect quantification results.

This misalignment is commonly addressed by the family of *Adjusted Classify & Count* (ACC) methods, which use an estimate of the classifier's confusion matrix to adjust the predicted label prevalences [35, 29, 8]. ACC has been shown to estimate the true test label prevalences in expectation if the so-called *prior probability shift* (PPS) assumption holds [33].

In this paper, we investigate ACC in the context of graph-structured data and describe why it is oftentimes ill-suited to tackle graph quantification problems. To solve this problem, we propose two novel methods for graph quantification learning: *Structural importance sampling* (SIS) and *Neighborhood-aware ACC* (NACC). First, SIS generalizes ACC to the (structural) covariate shift setting; to our knowledge, this is the first quantification method that tackles covariate shift systematically. Second, NACC further improves the quantification performance of ACC in the graph domain by improving class identifiability in the presence of non-homophilic edges. We begin with a brief formal description of the general quantification problem in Section 2. In Section 3 we then consider graph quantification and introduce our novel SIS and NACC methods. In Section 4, the proposed methods are evaluated on a series of datasets under different shift assumptions. Last, we conclude with a brief outlook in Section 5.

## 2   Quantification Learning

Let $\mathcal{X}$ denote the instance space and $\mathcal{Y} = \{1, \ldots, K\}$ the (finite) label space. In QL we assume to be given a training set of labeled instances $\mathcal{D}_L \subseteq \mathcal{X} \times \mathcal{Y}$ drawn from a distribution $P$ with corresponding density $p$. Additionally, there is a set of labeled instances $\mathcal{D}_U \subseteq \mathcal{X} \times \mathcal{Y}$ drawn from a test distribution $Q$ with corresponding density $q$. Let $X$ and $Y$ denote *random variable*s (RVs) that project the joint instance-label space to the instance and label spaces, respectively. The goal of QL is to estimate the pushforward measure $Q(Y)$ given samples $\mathcal{D}_L$ and $\mathcal{X}_U := \{x \mid (x, y) \in \mathcal{D}_U\}$. If $P = Q$, i.e., if the training and test data are drawn from the same distribution, the quantification problem is trivially solved via a maximum likelihood estimate of the label distribution on $\mathcal{D}_L$:

$$\hat{Q}^{\mathrm{MLPE}}(Y = i) := \frac{1}{|\mathcal{D}_L|} \sum_{(x,y) \in \mathcal{D}_L} \mathbb{1}[y = i] \tag{1}$$

where $\mathbb{1}[\cdot]$ denotes the indicator function. This *Maximum Likelihood Prevalence Estimation* (MLPE) approach [1, 5] is akin to the majority classifier in classification in the sense that it predicts the most likely distribution in the absence of test data $\mathcal{X}_U$. However, if the training and test data are not identically distributed, the quantification problem becomes more challenging. A quantification approach has to account for the distribution shift between $P$ and $Q$ to provide accurate estimates of $Q(Y)$. Depending on the nature of this distribution shift, different quantification methods may be more or less suitable.

### 2.1   Types of Distribution Shift

If the train and test distributions differ, one should ask whether learning from the training data is still feasible. Certainly, if $P$ and $Q$ are completely unrelated, any information learned from $\mathcal{D}_L$ is useless for predicting $Q(Y)$. Quantification approaches, therefore, typically assume that $P$ and $Q$ are related in some way. The applicability of a quantification method then depends on whether those assumptions hold true for the given problem. First, note that $Q$ can be expressed as

$$Q(X, Y) = Q(Y \mid X)Q(X) = Q(X \mid Y)Q(Y).$$

By fixing one of the factors in the two right-hand terms, we obtain three types of distribution shifts [5]:

1. **Concept Shift:** The conditional label distribution changes, but the distribution of the instances remains the same, i.e., $Q(Y \mid X) \neq P(Y \mid X)$, while $Q(X) = P(X)$. This type of shift, also referred to as *concept drift*, can occur in domains with classes that are defined relative to some frame of reference.

2. **Covariate Shift:** The distribution of the instances changes, but the conditional label distribution remains the same, i.e., $Q(X) \neq P(X)$, while $Q(Y \mid X) = P(Y \mid X)$. This is common in domain adaptation, where the training and test data are drawn from different but related domains. For example, assume the task is to predict the prevalence of a certain sentiment or opinion in social media posts. The training data may be drawn from one social media platform, while the test data is drawn from another. Given a post $x$, the probability of it expressing a certain sentiment $y$ is likely the same on both platforms, but the distribution of posts may differ.

3. **Prior Probability Shift:** The label distribution changes, but not the class-conditional instance distribution, i.e., $Q(Y) \neq P(Y)$, while $Q(X \mid Y) = P(X \mid Y)$. Similar to covariate shift, *prior probability shift* (PPS) occurs between domains that share the same label concepts. For example, consider the task of predicting the percentage of a population that has a certain disease. The training data may come from a case-control study consisting of an equal proportion of healthy and infected individuals, while the test data is drawn from the general population. Given $y \in \{infected, healthy\}$, the feature distribution of an individual $x$ should be the same between training and test, whereas the prevalence of the disease will likely not be.

We do not consider the case where $Q(Y) = P(Y)$, as this would imply that the label distribution remains unchanged, in which case the quantification problem is trivially solved by MLPE. Note that the difference between covariate shift and PPS is subtle. Whether it is $P(X)$ or $P(Y)$ that changes between training and test is mostly a matter of the assumed causal relation between instances and labels, i.e., whether it is in the direction $\mathcal{X} \to \mathcal{Y}$ or $\mathcal{Y} \to \mathcal{X}$ [6, 30, 18]. In QL, PPS is commonly assumed, as there are many $\mathcal{Y} \to \mathcal{X}$ domains in which this is reasonable [15]. Generally speaking, quantification under concept or covariate shift is more challenging and often requires additional assumptions or domain knowledge. We will get back to the question of which shift assumptions are appropriate for a given domain in Section 3.

## 2.2 Adjusted Count

We will now describe the *Adjusted Classify & Count* (ACC) method, a popular approach to quantification under PPS [8]. As mentioned in the introduction, the naive Classify & Count method (incorrectly) assumes that the predicted label prevalences of a classifier $h : \mathcal{X} \to \mathcal{Y}$ equal the true label prevalences, i.e., that $Q(\hat{Y}) = Q(Y)$, where $\hat{Y} = h(X)$ is a RV representing the predicted label. Under this assumption, the label prevalences can be estimated as

$$\hat{Q}^{\mathrm{CC}}(Y = i) := \hat{Q}(\hat{Y} = i) = \frac{1}{|\mathcal{X}_U|} \sum_{x \in \mathcal{X}_U} \mathbb{1}[h(x) = i] \quad \forall i \in \mathcal{Y} \, . \tag{2}$$

However, since $h$ is trained on data drawn from $P$, the estimated propensity scores $\hat{Q}^{\mathrm{CC}}(Y)$ will be biased towards $P(Y)$ in practice. ACC removes this bias by adjusting the predicted label prevalences based on an estimate of the classifier's confusion matrix. To understand ACC, note that the PPS assumption implies that for any measurable mapping $\phi : \mathcal{X} \to \mathcal{Z}$, we have $P(Z \mid Y) = Q(Z \mid Y)$ where $Z = \phi(X)$ [19]. This allows us to factorize $Q(Z)$ as follows:

$$Q(Z) = \sum_{i=1}^{K} Q(Z \mid Y = i)Q(Y = i) \overset{\mathrm{PPS}}{\Longleftrightarrow} Q(Z) = \sum_{i=1}^{K} P(Z \mid Y = i)Q(Y = i) \tag{3}$$

Let $\phi = h$, i.e., $Z = \hat{Y}$; then, we can plug the estimate of $Q(\hat{Y})$ from Eq. (2) and the estimate

$$\hat{Q}(\hat{Y} = j \mid Y = i) = \hat{P}(\hat{Y} = j \mid Y = i) = \frac{\sum\limits_{(x,y) \in \mathcal{D}_L} \mathbb{1}[h(x) = j \wedge y = i]}{|\{(x, y) \in \mathcal{D}_L \mid y = i\}|} \quad \forall i, j \in \mathcal{Y} \tag{4}$$

of $Q(\hat{Y} \mid Y)$ into Eq. (3). This yields a system of equations which can be solved to obtain estimates of $Q(Y)$ [29]. Let $\hat{\mathbf{C}} \in [0,1]^{K \times K}$ be the estimated confusion matrix of the classifier $h$ on $Q$, i.e., $\hat{\mathbf{C}}_{j,i} = \hat{Q}(\hat{Y} = j \mid Y = i)$. Then, the ACC estimates of $Q(Y)$ are given by

$$\hat{Q}^{\mathrm{ACC}}(Y) := \hat{\mathbf{C}}^{-1} \cdot \hat{Q}(\hat{Y}) . \tag{5}$$

While the binary version of ACC goes back at least to Gart and Buck [11], it was first described as a quantification method by Vucetic and Obradovic [35]. Tasche [33] showed that ACC is an unbiased estimator of the true test label prevalences if the PPS assumption holds.

Note that there are two practical problems with Eq. (5): First, if $\mathbf{C}$ is not invertible, there might be no or multiple solutions for $\hat{Q}^{\mathrm{ACC}}(Y)$. Second, the adjusted label prevalences may not be a valid distribution over $\mathcal{Y}$, i.e., they could lie outside $[0,1]$ or not sum to one. Possible reasons for this are that the PPS assumption might not be (fully) satisfied or simply that the estimates $\hat{\mathbf{C}}$ and $\hat{Q}(\hat{Y})$ are noisy, e.g., due to small sample sizes. A number of solutions to these problems have been proposed in the literature, including clipping and rescaling the estimates [10], adjusting the confusion matrix [19], using the pseudo-inverse of $\mathbf{C}$, or replacing the system of equations with a constrained optimization problem [3]. In this work, we will use the latter approach, i.e., constrained optimization, to solve Eq. (5):

$$\hat{Q}^{\mathrm{ACC}}(Y) = \operatorname*{arg\,min}_{\mathbf{q} \in \Delta_K} \left\| \hat{\mathbf{C}} \cdot \mathbf{q} - \hat{Q}(\hat{Y}) \right\|_2^2 , \tag{6}$$

where $\Delta_K$ denotes the unit $(K-1)$-simplex. This problem can be solved numerically, e.g., using a (quasi-)Newtonian method such as *Sequential Least Squares Quadratic Programming*. Bunse [3] has shown that this approach is a sensible default choice, as it generally performs well in practice.

In addition to the CC and ACC methods described above, which use a hard classifier $h : \mathcal{X} \to \mathcal{Y}$, one can also use a probabilistic classifier $h_s : \mathcal{X} \to \Delta_K$ [2]. Analogous to CC, *Probabilistic Classify & Count* (PCC) is defined as

$$\hat{Q}^{\mathrm{PCC}}(Y = i) := \frac{1}{|\mathcal{X}_U|} \sum_{x \in \mathcal{X}_U} h_s(x)_i . \tag{7}$$

Likewise, *Probabilistic Adjusted Classify & Count* (PACC) estimates $Q(\hat{Y})$ and $P(\hat{Y} \mid Y)$ using predicted label probabilities. The motivation for using a soft classifier instead of a hard one is that predicted label probabilities can be more informative than hard labels. Whether this is truly the case is problem-dependent and depends on the quality of the predicted probabilities.

## 3  Graph Quantification Learning

We now turn to the problem of quantification learning on graph-structured data. In Section 2, we assumed that the instances in $\mathcal{D}_L$ and $\mathcal{D}_U$ are i.i.d. wrt. $P$ and $Q$ respectively. This assumption does not hold for graph-structured data, where the instances are the vertices of a graph and the labels are associated with the vertices. More specifically, let $\mathcal{G} = (\mathcal{V}, \mathcal{E})$ be a graph with vertex set $\mathcal{V}$ and edge set $\mathcal{E} \subseteq \mathcal{V} \times \mathcal{V}$. Each vertex $v_i \in \mathcal{V}$ is associated with a feature vector $x_i \in \mathcal{X}$ and a label $y_i \in \mathcal{Y}$. We use $\mathcal{N}(v_i) = \{v_j \mid (v_i, v_j) \in \mathcal{E}\}$ to denote the set of neighbors of $v_i$. The edges in $\mathcal{G}$ are used to encode homophily between vertices, i.e., similar vertices are more likely to be connected. Formally, an edge $(v_i, v_j) \in \mathcal{E}$ should indicate that $P(y_i = y_j) \geq \varepsilon$, with $\varepsilon$ being either a graph-specific constant or a function of an edge weight $w_{i,j} \in \mathbb{R}$. Since homophily is symmetric by definition, $\mathcal{G}$ is undirected, i.e., $(v_i, v_j) \in \mathcal{E} \Leftrightarrow (v_j, v_i) \in \mathcal{E}$. Such homophilic graphs are commonly used to represent social networks, citation networks, co-purchase graphs, or the World Wide Web. Figure 1 shows one such graph, namely the Amazon Photos co-purchase graph [32], where vertices represent products, edges indicate that two products are frequently bought together, and labels represent product categories. Due to homophily, the product categories form separate densely connected clusters, while cross-category edges are sparse.

Analogous to the tabular case, in *graph quantification learning* (GQL) we are given a training set of labeled vertices $\mathcal{D}_L \subseteq \mathcal{V} \times \mathcal{Y}$ drawn from a distribution $P$ and our goal is to estimate the label distribution of the vertices in a test set $\mathcal{V}_U$ drawn from a distribution $Q(V)$, with $V$ denoting a RV mapping the joint measure space $\mathcal{V} \times \mathcal{Y}$ to $\mathcal{V}$. Given some vertex classifier $h : \mathcal{V} \to \mathcal{Y}$, the GQL

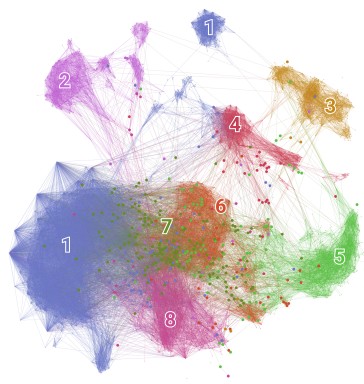

Figure 1: The Amazon Photos co-purchase graph. Colors indicate vertex labels ($K = 8$). The highlighted vertices are misclassifications by an APPNP classifier.

problem is, in principle, amenable to standard aggregative quantification methods, such as ACC or PACC. As discussed in Section 2.2, those adjusted count methods assume PPS, which in turn assumes a $\mathcal{Y} \to \mathcal{V}$ domain. This means that, both, the training and the test data are assumed to be generated by sampling from some fixed distribution $P(V \mid Y = i)$ for all $i \in \mathcal{Y}$. We argue that this is often not realistic for graph-structured data.

Consider the example of estimating the proportion of users holding a certain opinion. Here, the training data $\mathcal{D}_L$ may come from a social network where a (non-representative) local subset of users was sampled. The test data $\mathcal{D}_U$, on the other hand, may come from the entire social network or possibly some local subcluster of interest. In this setting, it is the instance distribution $P(V)$ that changes, while $P(Y \mid V)$ remains fixed, i.e., covariate shift. More generally, for a sampling process that is structure dependent, for example, by sampling local training or test neighborhoods, the covariate shift assumption is more appropriate than PPS. We will now discuss how such structural biases can be accounted for in the quantification process.

### 3.1 Structural Importance Sampling

ACC depends on being able to estimate the test confusion matrix $\mathbf{C}$ from training data. As described, this is trivial under PPS. We will now introduce *structural importance sampling* (SIS), a novel generalization of graph quantification learning to covariate shift. First, note that $\mathbf{C}_{j,i}$ can be expressed as

$$\mathbf{C}_{j,i} = \int_{v \in \mathcal{V}} \mathbb{1}[h(v) = j] dQ(V = v \mid Y = i) = \int_{v \in \mathcal{V}} \mathbb{1}[h(v) = j] \underbrace{\frac{q_{V|Y}(v \mid i)}{p_{V|Y}(v \mid i)}}_{= \rho_{V|Y}(v|i)} dP(V = v \mid Y = i) \quad (8)$$

with $q_{V|Y} = \frac{dQ(V|Y)}{d\mu}$ and $p_{V|Y} = \frac{dP(V|Y)}{d\mu}$ denoting the Radon-Nikodym derivatives of $Q$ and $P$ and $\mu$ a (counting) measure on $\mathcal{V}$, i.e., their *probability density functions* (PDFs), and $\rho_{V|Y}$ denoting the ratio between those PDFs. Using the covariate shift assumption, we get

$$\rho_{V|Y}(v \mid y) = \frac{q_{V|Y}(v \mid y)}{p_{V|Y}(v \mid y)} = \frac{q_{Y|V}(y \mid v) q_V(v) p_Y(y)}{p_{Y|V}(y \mid v) p_V(v) q_Y(y)} = \rho_V(v) \cdot \rho_Y(y)^{-1} . \quad (9)$$

Thus, $\mathbf{C}$ can be obtained by reweighting the vertices:

$$\mathbf{C}_{j,i} = \frac{\rho_Y(i)^{-1} \int_{\mathcal{V}} \mathbb{1}[h(v) = j] \rho_V(v) dP(V = v \mid Y = i)}{\rho_Y(i)^{-1} \int_{\mathcal{V}} \rho_V(v) dP(V = v \mid Y = i)} = \frac{\mathbb{E}_{P(V|Y=i)}[\mathbb{1}[\hat{Y} = j] \rho_V(V)]}{\mathbb{E}_{P(V|Y=i)}[\rho_V(V)]} \quad (10)$$

Given $\mathcal{D}_L$, we can obtain an unbiased estimate of $\mathbf{C}_{j,i} = Q(\hat{Y} = j \mid Y = i)$:

$$\hat{Q}(\hat{Y} = j \mid Y = i) = \frac{\displaystyle\sum_{(v,y) \in \mathcal{D}_L} \mathbb{1}[h(v) = j \wedge y = i] \cdot \rho_V(v)}{\displaystyle\sum_{(v,y) \in \mathcal{D}_L} \mathbb{1}[y = i] \cdot \rho_V(v)} . \quad (11)$$

Note that this is essentially a weighted version of Eq. (4). The problem with this formulation is that it requires $\rho_V = \frac{q_V}{p_V}$, which cannot be computed since, both, $P(V)$ and $Q(V)$ are unknown. We do, however, have access to samples from both distributions, i.e., $\mathcal{D}_L$ and $\mathcal{V}_U$. Using suitable vertex kernels $k_q, k_p : V \times V \to \mathbb{R}$, we can thus obtain estimate of the PDFs via kernel density estimation:

$$\hat{q}_V(v) = \frac{1}{|\mathcal{V}_U|} \sum_{v' \in \mathcal{V}_U} k_q(v, v') \quad \text{and} \quad \hat{p}_V(v) = \frac{1}{|\mathcal{D}_L|} \sum_{(v', y') \in \mathcal{D}_L} k_p(v, v') . \tag{12}$$

The suitability of the kernels depends on the nature of the distribution shift. Intuitively, $k_q(v, v')$ and $k_p(v, v')$ should be proportional to the probability of sampling a vertex $v$ from $Q(V)$ and $P(V)$, respectively, given that $v'$ has been sampled. For example, the constant kernel $k_1(v, v') = 1$ describes a sampling process where the probability of sampling $v$ is independent of $v'$. Using $k_q = k_p = k_1$, Eq. (11) simplifies to standard ACC (cf. Eq. (4)), i.e., SIS is a generalization of ACC.

Under (structural) covariate shift, $k_q$ and $k_p$ should be chosen in accordance with the sampling process. Since the family of structural shifts is broad, there is no single, generally applicable kernel. Nonetheless, if the shift is induced by a localized sampling process where vertices are sampled via *random walk*s (RWs), the probability of sampling $v$ given $v'$ is proportional to the number of RWs between both vertices. This probability can be computed via the *personalized page-rank* (PPR) algorithm [26]:

$$k_{\text{PPR}}(v_i, v_j) = \Pi_{i,j}, \quad \text{where} \quad \Pi = \left( \alpha \mathbf{I} + (1 - \alpha) \bar{\mathbf{A}} \right)^L . \tag{13}$$

Here, $\bar{\mathbf{A}} = \mathbf{A} \mathbf{D}^{-1}$ is the normalized adjacency matrix of the graph, $\alpha \in (0, 1)$ is a teleportation parameter and $L$ is the number of steps in the random walk. We found that $k_{\text{PPR}}$ is a good default for localized structural shifts. Nonetheless, depending on the problem domain, other choices might be more appropriate. A more in-depth discussion of the kernel selection can be found in Appendix B.1. A formal analysis of the computational complexity of SIS under different kernel choices is provided in Appendices C.1 and C.2.

To summarize, SIS enables graph quantification under covariate shift by estimating $Q(\hat{Y} \mid Y)$ using kernel density estimates of the vertex distributions $P(V)$ and $Q(V)$. Using these estimates, the adjusted label prevalences can be computed using Eq. (6).

### 3.2 Neighborhood-aware Adjusted Count

In the previous section, we extended GQL to covariate shift. Next, we will address another orthogonal problem of ACC: *Class identifiability*. Consider a classifier $h$ that is unable to distinguish between two classes $i$ and $j$, i.e., it predicts the same label for both. In this case, $\mathbf{C}_{:,i} = \mathbf{C}_{:,j}$; thus, there is no unique solution for Eq. (5). This can lead to poor quantification results if the prediction vector $Q(\hat{Y})$ has a large overlap with, both, $\mathbf{C}_{:,i}$ and $\mathbf{C}_{:,j}$, since any distribution of probability mass between both classes may then be returned. To address this issue, we propose *Neighborhood-aware ACC* (NACC), which uses the neighborhood structure of the graph to improve class identifiability.

First, note that Eq. (6) can be understood as finding a mixture of the columns of $\mathbf{C}$ that best approximates $Q(\hat{Y})$. In the case of collinear columns, this mixture is not unique. A simple way to break such symmetries is to set $Z = \hat{Y}_{\mathcal{N}}$ in Eq. (3), where $\hat{Y}_{\mathcal{N}}$ is a RV representing a tuple of the predicted label of a vertex and the majority predicted label of its neighbors:

$$Q(\hat{Y}_{\mathcal{N}} = (j, k)) = \sum_{i=1}^{K} Q(\hat{Y}_{\mathcal{N}} = (j, k) \mid Y = i) \cdot Q(Y = i) .$$

Using this decomposition of $Q(\hat{Y}_{\mathcal{N}})$, $Q(Y)$ can be estimated using ACC and, possibly, SIS. Intuitively, this approach uses homophily information to improve class identifiability. Consider Fig. 1, where a vertex is highlighted if it is misclassified by an *approximate personalized propagation of neural predictions* (APPNP) classifier [12]. Note that the vertices with label 7 (dark green) are often confused with vertices of label 1 (blue) or 6 (orange) because there are many non-homophilic edges between those classes. Using ACC, this would imply that the row vectors of labels 7, 1 and 6 are collinear, i.e., $C_{:,7} \approx \alpha \cdot C_{:,1} + (1 - \alpha) \cdot C_{:,6}$ for some $\alpha \in [0, 1]$. Using the neighborhood structure, NACC can break this symmetry. For labels 1 and 6, the majority of predicted neighbors will

nearly always be of the same label due to homophily, whereas for label 7, both, $\hat{Y}_{\mathcal{N}} = (1, 6)$ and $\hat{Y}_{\mathcal{N}} = (6, 1)$ are common. With this information, NACC is able to distinguish the confusion profile of label 7 from those of labels 1 and 6.

In principle, one could extend NACC to use even more neighborhood information, e.g., by considering the majority label of the neighbors of neighbors or by considering the second-most predicted neighboring label. However, given a finite training set $\mathcal{D}_L$, by making the confusion profiles more fine-grained, the confusion estimate $\hat{\mathbf{C}}$ will become noisier, counteracting the potential gains of additional information. We found that using the 1-hop majority label is a good trade-off between class identifiability and confusion estimate noise. An analysis of the computational complexity of NACC can be found in Appendix C.3.

## 4 Evaluation

We assess the performance of SIS and NACC on a series of graph quantification tasks using, both, PPS and covariate shift. The quantifiers are applied to the predictions of a set of node classifiers. As a baseline we compare our proposed GQL methods with MLPE, (P)CC and (P)ACC. We build upon the `QuaPy` Python library (BSD 3-Clause). Further details can be found in Appendix A.

### 4.1 Experimental Setup

**Quantification Metrics**  There is a large number of metrics to evaluate quantification methods [5]. We use *Absolute error* (AE) and *relative absolute error* (RAE):

$$\text{AE}(q, \hat{q}) = \frac{1}{K} \sum_{i=1}^{K} |q_i - \hat{q}_i| \qquad\qquad \text{RAE}(q, \hat{q}) = \frac{1}{K} \sum_{i=1}^{K} \frac{|q_i - \hat{q}_i|}{q_i} \qquad (14)$$

AE penalized all errors equally, whereas RAE [16] penalizes errors on rare labels more heavily.

**Datasets**  Since the literature on GQL is scarce, there are no established benchmark datasets for this task. For this reason, we synthetically generate quantification tasks from the following five node classification datasets: 1. CoraML, 2. CiteSeer, 3. PubMed, 4. Amazon Photos and 5. Amazon Computers [21, 14, 13, 31, 25, 20, 32]. The first three datasets are citation networks, where the nodes are documents and the edges represent citations between them. The two Amazon datasets are product co-purchasing graphs, where the nodes are products and the edges represent that are often bought together. All nodes are labeled with the topic or product category they belong to. All reported results were obtained by averaging over 10 random splits of the node set into classifier-train/quantifier-train/test, with sizes of $5\%/15\%/80\%$ respectively.

Additionally, we conduct a study on the "Twitch Gamers" dataset [28]; it consists of about 168k vertices and 6.8 million edges, where vertices represent Twitch accounts and edges represent followership relations. Each vertex is annotated with the language of the corresponding user, whether the user streams explicit content, and a number of other features. This social network dataset is a good source of real-world structural distribution shifts, since, for example, users from the same language community tend to form more densely connected subgraphs than users from different language communities (see Fig. 2). We select 10% of the users uniformly at random as classifier-train nodes with a binary target label indicating whether a given user streams explicit content. From the remaining nodes, another 10% are selected as quantifier-train nodes, i.e., as $\mathcal{D}_L$. Last, the then remaining nodes are partitioned into top-5 languages spoken by the users: English (74%), German (5%), French (4%), Spanish (3%) and Russian (2%); users outside of those languages are discarded. For each of the five partitions, the (binary) quantification task is to estimate the prevalence of explicit content streamers.

**Distribution Shift**  For the synthetic quantification experiments, we introduce shifts in the test partitions, while the training data is sampled uniformly at random from the training split. We consider three types of test distribution shifts:

1. PPS: To simulate PPS, we first sample $10 \cdot K$ target label distribution $q \in \Delta_K$ from a Zipf distribution over the labels [27]. For each sampled target distribution, we then sample 100 vertices such that the target label frequencies are reached.

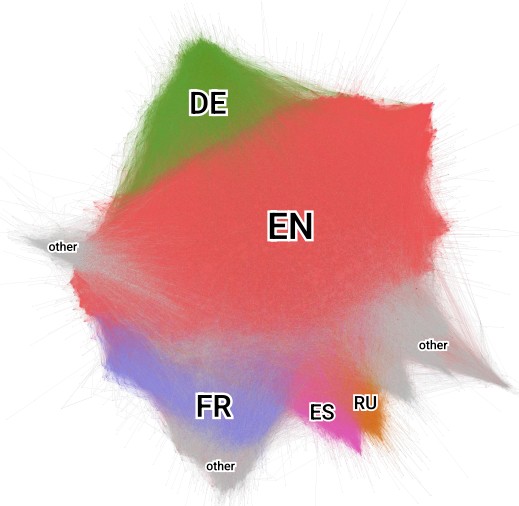

Figure 2: Visualization of the "Twitch Gamers" dataset [28]. Vertices represent Twitch users, edges represent follower relationships. Colors indicate the primary language of each user.

2. Structural covariate shift via *breadth-first search* (BFS): For each label, we select 10 corresponding vertices and starting at each of those vertices we sample 100 nodes via BFS.

3. Structural covariate shift via RWs: Analogous to the BFS setting, we sample 100 nodes via random walks of length 10 with teleportation parameter $\alpha = 0.1$.

For the real-world "Twitch Gamers" dataset, no synthetic distribution shifts are introduced; instead we use the natural covariate shift occurring when partitioning users by language. This allows us to compare the applicability of the shift assumptions made by different quantification approaches.

**Classifiers** We use four different node classifiers to predict the labels of the vertices: A structure-unaware *Multilayer Perceptron (MLP)*, a *Graph Convolutional Network (GCN)* [17], *Graph Attention Network (GAT)* [34], and *APPNP* [12]. All models are trained using the same training splits and hyperparameters, and two hidden layers/convolutions with widths of 64 and ReLU activations. Each model is trained ten times on each of the ten splits per dataset, totalling 100 models per dataset, with which each quantifier is evaluated. Additionally, we evaluate the previously proposed *Ego-network Quantification* (ENQ) GQL method by Milli et al. [22]. ENQ uses a simple neighborhood majority classifier combined with (standard) ACC; in addition, we evaluate it using NACC and SIS.

**Quantifiers** We evaluate SIS and NACC and compare them against standard (P)ACC and (P)CC. For SIS, we use $k_q = k_\lambda$, with $k_\lambda$ being an interpolated version of the PPR kernel from Eq. (13):

$$k_\lambda(v, v') = \lambda k_{\text{PPR}}(v, v') + (1 - \lambda),$$

where $\lambda \in [0, 1]$ is a hyperparameter that controls the minimum weight that should be assigned to each vertex. In the following, we report results for $\lambda = 0.9$; evaluations with different $\lambda$ parameterizations and an alternative shortest-path-based vertex kernel can be found in Appendix B. For the *kernel density estimation* (KDE) estimate of $p_V$, we use the constant kernel $k_p = k_1$, since the training data is sampled uniformly at random without being subject to synthetic distribution shift in our setup. This implies $\rho_V = q_V$, simplifying the SIS estimation.

### 4.2 Discussion of Synthetic Distribution Shift Results

Table 1 compares the quantification performance of PCC, PACC and our extensions of the latter, i.e., SIS and neighborhood-aware PACC under synthetically induced distribution shifts. Additionally, the last block of columns shows the average rank of each quantifier across all datasets. **Bold numbers** indicate that there is no statistically significant difference between the reported mean and the best mean within a given block, determined by the 95th percentile of a one-sided t-test. Standard errors are not reported for visual clarity and space reasons, since they are mostly very close to zero.

Table 1: Quantification using probabilistic classifiers (absolute error and relative absolute error).

| Model & Shift | Quantifier | CoraML AE | RAE | CiteSeer AE | RAE | A. Photos AE | RAE | A. Computers AE | RAE | PubMed AE | RAE | Avg. Rank AE | RAE |
|---|---|---|---|---|---|---|---|---|---|---|---|---|---|
| PPS | MLPE | .0903 | .5692 | .0469 | .2623 | .0924 | .6660 | .0770 | .5358 | .1268 | .3948 | - | - |
| EgoNet PPS | PCC | .0750 | .8692 | **.0429** | **.3543** | .0480 | 1.319 | .0453 | .5297 | .1159 | .4764 | 3.8 | 4.2 |
| | PACC | .0848 | .6863 | .0927 | .5928 | .0314 | .3881 | .0334 | .3437 | .0833 | .3222 | 4.4 | 4.0 |
| | NEIGH PACC | **.0519** | **.4397** | .0594 | **.3791** | .0227 | .3265 | **.0258** | **.2693** | .0373 | **.1376** | 2.2 | 2.0 |
| | SIS PACC | .0837 | .6832 | .0916 | .5871 | .0313 | .3925 | .0333 | .3433 | .0817 | .3183 | 3.4 | 3.4 |
| | SIS NEIGH PACC | **.0514** | **.4369** | .0591 | **.3775** | **.0227** | **.3330** | **.0258** | **.2691** | .0371 | **.1374** | 1.2 | 1.4 |
| MLP PPS | PCC | .0827 | .8565 | .0361 | .2782 | .0497 | 1.105 | .0533 | .6342 | .0470 | .1870 | 5.0 | 5.0 |
| | PACC | .0481 | .4186 | .0336 | .2271 | .0191 | .3036 | .0334 | .3690 | **.0181** | .0649 | 3.2 | 3.2 |
| | NEIGH PACC | **.0326** | **.2865** | .0288 | .1908 | **.0163** | .3595 | **.0265** | **.2936** | .0187 | .0649 | 2.4 | 2.6 |
| | SIS PACC | .0486 | .4237 | .0327 | .2218 | .0192 | **.2881** | .0338 | .3708 | .0179 | **.0641** | 3.4 | 2.8 |
| | SIS NEIGH PACC | **.0320** | **.2847** | **.0271** | **.1799** | **.0162** | .3345 | **.0263** | **.2913** | .0178 | **.0616** | 1.0 | 1.4 |
| GAT PPS | PCC | .0479 | .5323 | .0219 | .1573 | .0314 | .9570 | .0398 | .4674 | .0463 | .1911 | 5.0 | 5.0 |
| | PACC | .0297 | .2660 | .0192 | .1262 | **.0147** | **.2776** | .0217 | .2326 | .0176 | .0635 | 3.2 | 2.8 |
| | NEIGH PACC | **.0287** | **.2438** | .0199 | .1307 | **.0146** | .3249 | **.0211** | **.2271** | .0192 | .0694 | 2.6 | 3.0 |
| | SIS PACC | **.0290** | .2635 | **.0181** | **.1200** | .0148 | .2901 | .0214 | .2299 | .0167 | **.0606** | 2.4 | **2.0** |
| | SIS NEIGH PACC | **.0279** | **.2414** | .0187 | .1235 | **.0146** | .3466 | **.0207** | **.2238** | .0181 | .0655 | 1.8 | 2.2 |
| GCN PPS | PCC | .0438 | .4697 | .0221 | .1574 | .0315 | .8508 | .0391 | .4667 | .0405 | .1665 | 5.0 | 5.0 |
| | PACC | .0246 | .2216 | .0190 | .1259 | **.0122** | **.2056** | .0228 | .2411 | .0161 | .0591 | 3.0 | 3.0 |
| | NEIGH PACC | .0239 | **.2073** | .0188 | .1253 | .0134 | .2920 | **.0191** | **.2054** | .0181 | .0659 | 3.0 | 2.6 |
| | SIS PACC | **.0234** | .2163 | **.0178** | **.1186** | **.0124** | .2295 | .0223 | .2386 | **.0151** | **.0555** | 2.0 | **2.2** |
| | SIS NEIGH PACC | **.0232** | **.2073** | **.0176** | **.1177** | .0135 | .3329 | **.0188** | **.2029** | .0168 | .0613 | 2.0 | **2.2** |
| APPNP PPS | PCC | .0374 | .4124 | .0214 | .1509 | .0318 | .9795 | .0390 | .4657 | .0398 | .1664 | 5.0 | 5.0 |
| | PACC | .0217 | .1986 | .0184 | .1211 | **.0124** | **.2442** | .0256 | .2638 | .0165 | .0597 | 2.6 | 2.8 |
| | NEIGH PACC | .0224 | **.1943** | .0184 | .1222 | .0139 | .3133 | **.0231** | **.2471** | .0187 | .0676 | 3.4 | 3.2 |
| | SIS PACC | **.0203** | **.1926** | **.0171** | **.1132** | .0130 | .3058 | .0249 | .2566 | **.0154** | **.0558** | 1.6 | 1.6 |
| | SIS NEIGH PACC | .0214 | **.1939** | .0172 | .1149 | .0143 | .3780 | **.0226** | **.2424** | .0175 | .0642 | 2.4 | 2.4 |
| BFS | MLPE | .1839 | 8.185 | .2676 | 15.22 | .1622 | 9.313 | .1180 | 5.666 | .3034 | 5.067 | - | - |
| EgoNet BFS | PCC | .1440 | 8.886 | .2374 | 22.27 | .0566 | 3.686 | .0455 | 2.274 | .2313 | 26.15 | 5.0 | 5.0 |
| | PACC | .1005 | **5.140** | .1908 | 17.87 | **.0388** | **1.112** | .0419 | 1.190 | .2108 | 24.61 | 3.6 | 2.6 |
| | NEIGH PACC | **.0825** | **4.300** | **.1552** | **14.32** | .0359 | 1.151 | **.0384** | 1.156 | .1605 | 18.38 | 2.0 | 2.0 |
| | SIS PACC | .1003 | **5.196** | .1909 | 17.93 | **.0385** | 1.125 | .0413 | 1.192 | .2119 | 24.96 | 3.4 | 3.6 |
| | SIS NEIGH PACC | **.0813** | **4.244** | **.1531** | **14.12** | .0357 | 1.165 | **.0379** | 1.162 | .1603 | 18.37 | 1.0 | 1.8 |
| MLP BFS | PCC | .1243 | 7.212 | .1588 | 14.84 | .0668 | 4.028 | .0662 | 3.635 | **.0800** | 10.44 | 4.4 | 5.0 |
| | PACC | .0645 | 3.508 | .1158 | 10.63 | .0237 | .9928 | .0392 | 1.608 | **.0816** | 7.663 | 3.0 | 2.8 |
| | NEIGH PACC | **.0577** | **3.008** | **.0984** | **8.845** | .0290 | 1.237 | .0400 | 1.817 | .0878 | **6.787** | 3.4 | 2.6 |
| | SIS PACC | .0637 | 3.461 | .1162 | 10.74 | **.0222** | **.9079** | **.0370** | 1.509 | .0786 | 7.827 | **2.0** | 2.6 |
| | SIS NEIGH PACC | **.0560** | **2.972** | **.0964** | **8.699** | .0266 | 1.097 | **.0375** | 1.694 | .0840 | **6.833** | 2.2 | **2.0** |
| GAT BFS | PCC | .0741 | 4.840 | .0820 | 7.349 | .0291 | 1.757 | .0455 | 2.415 | **.0650** | 9.922 | 4.2 | 5.0 |
| | PACC | .0561 | 2.533 | .0656 | 5.347 | .0243 | **.7255** | .0331 | **.9463** | .0930 | **6.906** | 2.6 | 2.4 |
| | NEIGH PACC | .0577 | 2.548 | .0702 | 5.886 | .0280 | .8623 | .0363 | 1.234 | .1011 | 8.096 | 4.2 | 4.0 |
| | SIS PACC | **.0500** | **2.265** | **.0616** | **5.015** | .0226 | **.7147** | **.0312** | **.9381** | .0891 | 6.881 | 1.2 | **1.2** |
| | SIS NEIGH PACC | **.0502** | **2.179** | **.0621** | 5.134 | .0264 | .8485 | .0343 | 1.223 | .0962 | 7.892 | 2.8 | 2.4 |
| GCN BFS | PCC | .0539 | 3.489 | .0783 | 7.060 | .0256 | 1.513 | .0418 | 2.255 | **.0573** | 9.553 | 4.0 | 5.0 |
| | PACC | .0488 | 2.093 | .0637 | 5.267 | .0241 | **.5966** | .0401 | **.9320** | .0888 | **6.713** | 3.4 | 2.2 |
| | NEIGH PACC | .0474 | 2.020 | .0653 | 5.428 | .0261 | .6773 | .0379 | .9846 | .0977 | 7.994 | 4.0 | 3.4 |
| | SIS PACC | **.0415** | **1.943** | **.0618** | 5.132 | **.0207** | **.5932** | **.0358** | **.9569** | .0840 | **6.727** | 1.8 | 1.8 |
| | SIS NEIGH PACC | **.0402** | **1.786** | **.0595** | **4.910** | .0240 | .7070 | **.0351** | 1.004 | .0924 | 7.723 | 1.8 | 2.6 |
| APPNP BFS | PCC | .0469 | 3.074 | .0737 | 6.609 | .0271 | 1.492 | .0468 | 2.339 | **.0569** | 9.867 | 4.2 | 5.0 |
| | PACC | .0457 | 1.881 | .0603 | 4.944 | .0225 | **.5731** | .0430 | **.9227** | .0927 | **7.449** | 2.8 | 2.0 |
| | NEIGH PACC | .0459 | 1.910 | .0633 | 5.243 | .0260 | .6585 | .0435 | .9606 | .1017 | 8.673 | 4.2 | 3.6 |
| | SIS PACC | **.0380** | **1.729** | **.0574** | **4.705** | **.0213** | **.5823** | **.0395** | **.9331** | .0874 | **7.372** | 1.6 | **1.8** |
| | SIS NEIGH PACC | **.0378** | **1.615** | **.0562** | **4.600** | .0256 | .6993 | .0406 | **.9747** | .0962 | 8.440 | 2.2 | 2.6 |
| RW | MLPE | .1832 | 6.430 | .2651 | 15.07 | .1594 | 8.278 | .1158 | 4.466 | .3025 | 3.073 | - | - |
| EgoNet RW | PCC | .1355 | 6.275 | .2237 | 21.00 | .0538 | 2.895 | **.0449** | 1.703 | .2171 | 4.937 | 4.6 | 5.0 |
| | PACC | .0973 | **2.916** | **.1477** | **13.48** | .0476 | **1.061** | .0481 | **1.052** | .1506 | **1.716** | 3.8 | **1.2** |
| | NEIGH PACC | **.0900** | 3.201 | **.1476** | 13.49 | **.0418** | 1.165 | **.0427** | 1.091 | .1419 | 1.996 | 2.0 | 3.2 |
| | SIS PACC | .0971 | 2.939 | **.1484** | 13.55 | .0474 | 1.076 | .0478 | 1.058 | .1508 | 1.756 | 3.6 | 2.4 |
| | SIS NEIGH PACC | **.0887** | 3.101 | **.1455** | **13.29** | **.0416** | 1.173 | **.0424** | 1.100 | .1416 | 2.003 | 1.0 | 3.2 |
| MLP RW | PCC | .1263 | 5.275 | .1494 | 13.84 | .0727 | 3.820 | .0718 | 3.224 | .0913 | 1.376 | 4.8 | 5.0 |
| | PACC | .0733 | 2.347 | .0869 | 7.425 | .0332 | 1.251 | .0471 | 1.837 | **.0882** | **.7452** | 3.4 | 3.0 |
| | NEIGH PACC | .0644 | 2.153 | **.0824** | **7.105** | .0326 | 1.369 | .0434 | 1.859 | .0915 | **.7066** | 2.8 | 2.8 |
| | SIS PACC | .0743 | 2.417 | .0899 | 7.766 | .0315 | **1.144** | .0450 | 1.765 | .0864 | .7580 | 2.8 | 3.0 |
| | SIS NEIGH PACC | **.0620** | **2.040** | **.0798** | **6.894** | .0303 | 1.201 | **.0406** | 1.735 | .0880 | **.6960** | 1.2 | 1.2 |
| GAT RW | PCC | .0799 | 3.555 | .0766 | 6.693 | .0340 | 1.689 | .0500 | 2.195 | **.0691** | .7488 | 4.2 | 5.0 |
| | PACC | .0610 | 1.648 | .0594 | 4.563 | .0293 | **.7767** | .0382 | **.9639** | .0952 | **.6084** | 3.2 | 2.2 |
| | NEIGH PACC | .0617 | 1.836 | .0644 | 5.184 | .0308 | .9098 | .0394 | 1.191 | .0977 | .6290 | 4.2 | 4.0 |
| | SIS PACC | **.0550** | **1.427** | **.0554** | **4.203** | **.0277** | **.7664** | **.0366** | **.9656** | .0930 | **.6001** | 1.4 | **1.2** |
| | SIS NEIGH PACC | **.0540** | 1.496 | **.0564** | 4.432 | .0290 | .8826 | .0376 | 1.185 | .0951 | .6238 | 2.0 | 2.6 |
| GCN RW | PCC | .0539 | 2.085 | .0694 | 5.990 | .0276 | 1.247 | .0451 | 1.961 | **.0566** | **.4972** | 3.6 | 4.2 |
| | PACC | .0571 | 1.267 | .0554 | **4.204** | .0298 | **.6101** | .0428 | **.8952** | .0956 | .5915 | 4.2 | 2.4 |
| | NEIGH PACC | .0541 | 1.311 | .0588 | 4.637 | .0276 | **.6622** | .0403 | .9491 | .0964 | .6260 | 3.8 | 3.8 |
| | SIS PACC | .0494 | **1.089** | **.0527** | 4.001 | .0266 | **.5993** | .0392 | .9029 | .0925 | .5778 | 2.0 | **1.6** |
| | SIS NEIGH PACC | **.0461** | **1.034** | **.0524** | 4.049 | .0256 | .6682 | **.0375** | .9611 | .0934 | .6161 | 1.4 | 3.0 |
| APPNP RW | PCC | .0465 | 1.750 | .0659 | 5.638 | .0293 | 1.197 | .0504 | 2.016 | **.0546** | **.4160** | 3.8 | 4.2 |
| | PACC | .0527 | 1.121 | .0541 | 4.060 | .0282 | **.5726** | .0452 | **.8693** | .0979 | .5958 | 3.8 | 2.4 |
| | NEIGH PACC | .0512 | 1.229 | .0583 | 4.580 | .0276 | .6387 | .0456 | .9643 | .0999 | .6302 | 4.0 | 3.8 |
| | SIS PACC | .0448 | **.9489** | **.0501** | **3.735** | **.0255** | **.5650** | .0418 | **.8731** | .0944 | .5821 | 1.4 | **1.6** |
| | SIS NEIGH PACC | **.0425** | **.8951** | **.0508** | **3.873** | .0261 | .6387 | **.0424** | .9700 | .0965 | .6157 | 2.0 | 3.0 |

Table 2: Quantification results on the "Twitch Gamers" dataset with real-world covariate shift.

| Model | Quantifier | English | | German | | French | | Spanish | | Russian | | Avg. Rank | |
|---|---|---|---|---|---|---|---|---|---|---|---|---|---|
| | | AE | RAE | AE | RAE | AE | RAE | AE | RAE | AE | RAE | AE | RAE |
| **MLP** | CC | .014 | .028 | .120 | .257 | .115 | .246 | .210 | .512 | .205 | .539 | 2.8 | 2.8 |
| | ACC | .030 | .060 | **.002** | **.004** | .094 | .201 | .254 | .619 | **.066** | **.172** | 2.2 | 2.2 |
| | SIS ACC | .023 | .047 | .013 | .028 | .107 | .228 | .262 | .639 | .080 | .211 | 2.8 | 2.8 |
| | SIS NEIGH ACC | **.008** | **.016** | .138 | .297 | **.032** | **.069** | **.168** | **.408** | .207 | .544 | 2.2 | 2.2 |
| **APPNP** | CC | **.004** | **.008** | .119 | .255 | .115 | .246 | .210 | .512 | .177 | .465 | 2.4 | 2.4 |
| | ACC | .021 | .042 | .011 | .023 | **.089** | **.190** | .244 | .595 | .089 | .233 | 2.4 | 2.4 |
| | SIS ACC | .011 | .021 | **.008** | **.016** | .090 | .193 | .241 | .587 | **.080** | **.210** | 1.8 | 1.8 |
| | SIS NEIGH ACC | .096 | .191 | .399 | .858 | .177 | .379 | **.040** | **.097** | .256 | .671 | 3.4 | 3.4 |

In summary, considering the average ranks, our experiments show that both SIS and NACC are able to improve quantification results under PPS *and* covariate shift. The results are consistent across all classifiers, quantifiers, and types of distribution shifts, with either SIS or the combination of SIS and NACC performing best.

**Influence of the Classifier**    Unsurprisingly, the choice of classifier has a significant impact on the quantification performance. Even though a good classifier $h$ is not required by QL to obtain an unbiased estimate of the label prevalences, the quality of this estimate is still correlated with the classifier's accuracy. Overall, the naive neighborhood-based ENQ performs worst, followed by the structure-unaware MLP, while APPNP performs best.

**Influence of the Type of Distribution Shift**    The SIS kernels used in the experiments are based on the assumption that the distribution shift is induced by sampling localized random walks. In the RW covariate setting, this assumption is satisfied by definition, while in the BFS setting, the PPR kernel does, at least in theory, not fully capture the underlying sampling behavior. Nonetheless, SIS is able to improve quantification results in both cases. Interestingly, even in the PPS setting, where SIS is not necessary to account for the shift, we observe a clear improvement over ACC.

Experimental results for different kernel choices and hyperparameter settings can be found in Appendix B. Additionally, in Appendix D, we demonstrate the importance of using a structural kernel by comparing against a feature-based variant of SIS that does not consider structural information.

### 4.3   Discussion of Real-world Covariate Shift Results

Table 2 shows the quantification results on the real-world "Twitch Gamers" dataset. Here, the advantage of SIS and/or NACC over standard ACC and CC is less pronounced compared to the synthetic experiments. For the English language community, which also constitutes 74% of the training data, SIS performs worse than standard approaches – likely because there is only a small distribution shift compared to the training data for this community. However, for the other language communities, which are more strongly affected by distribution shift, SIS generally outperforms standard ACC and CC given an APPNP classifier. Using a weaker, structure-unaware MLP classifier, SIS is not able to significantly improve quantification results. This highlights the importance of a sufficiently accurate base classifier for effective aggregative quantification.

## 5   Conclusion

We have introduced two novel graph quantification methods, SIS and NACC; to our knowledge, this is the first work to investigate classifier-based graph quantification and the (structural) covariate shift problem. SIS enables quantification under covariate shift via kernel density estimates of the instance distributions. NACC uses the neighborhood structure of the graph to improve class identifiability. The effectiveness of our approach was demonstrated on multiple graph benchmark datasets.

We envision two lines of future research. First, in this work, we focused on extensions of ACC to the graph setting. Another family of methods are the so-called distribution matching quantifiers, e.g., DMy [16] or KDEy [24]. An extension of distribution matching approaches to GQL would be interesting. Second, there are no true graph quantification benchmark datasets currently, which is why we resorted to introducing synthetic dataset shifts to node classification benchmarks. Creating a true graph quantification dataset, e.g., using social media data, would be a valuable next step to assess the performance of SIS and NACC in the real-world.

## Acknowledgments and Disclosure of Funding

We gratefully acknowledge the support of the Munich Center for Machine Learning (MCML), and the German Research Center for Artificial Intelligence (DFKI). We also thank the reviewers for their valuable and constructive feedback, which helped to improve our work.

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

# Appendix for "Adjusted Count Quantification Learning on Graphs"

## A  Further Details on the Implementation

In our implementation, we use `torch-geometric` [7] (MIT license) for the *graph neural network* (GNN) models, while the `QuaPy` Python library (BSD 3-Clause) was used for the quantification methods. Additionally, we used Nvidia's `cuGraph` library (Apache 2.0 license) for GPU-based graph traversal and distance computation, e.g., to create BFS-based covariate shift. All experiments were conducted on a single machine with an AMD Ryzen 9 5950X CPU, 64GB RAM and an Nvidia RTX 4090 GPU with 24GB VRAM. Our code is available at `https://github.com/Cortys/graph-quantification`; it includes a versioned list of all dependencies that were used.

## B  Kernel Selection & Influence of Kernel Hyperparameters

In Section 4.1, we defined the interpolated PPR kernel as

$$k_\lambda(v, v') = \lambda k_{\mathrm{PPR}}(v, v') + (1 - \lambda) . \tag{15}$$

All reported results for SIS use this kernel with $\lambda = 0.9$. In this section, we will explain why this particular kernel was chosen.

### B.1  On Kernel Selection for SIS

As described in Section 3.1, SIS uses two vertex kernels, $k_p$ and $k_q$, to estimate the densities $p_V$ and $q_V$ of the training and test vertices, respectively. Depending on the type of distribution shift and other available domain knowledge, different kernels can be used. Overall, we found that the PPR kernel is a good choice for structural covariate shift.

Recall that the kernels are used to estimate the density ratio $\rho_V(v)$ for each training vertex $v$. Those density ratios are then used to reweight the training vertices in the confusion matrix estimate $\hat{\mathbf{C}}_{j,i}$, i.e., a high weight is assigned to training vertices from $\mathcal{D}_L$ that "look like" they have been sampled from the same distribution $Q(V)$ as the unlabeled test vertices $\mathcal{V}_U$ (see Eq. (11)). Assuming structural covariate shift between $P$ and $Q$, the probability of sampling a vertex $v$ given that $v'$ was sampled should depend on some measure of the structural distance between $v$ and $v'$. Since there are many different ways to define such a vertex distance (*shortest path*, random walk, graph spectrum, etc.), there are many potential SIS kernels.

If we do not have any prior knowledge about the distribution shift, other than that it is structural and that the underlying graph is homophilic, the problem of estimating the probability of sampling a vertex $v$ given that $v'$ was sampled is ill-posed. Fortunately, we do not need to derive a perfect kernel to obtain good quantification results. Instead, note that:

1. The kernel-based estimates are used to reweight the training vertices in the confusion matrix estimate $\hat{\mathbf{C}}_{j,i}$ such that vertices that are *close* to the test distribution have a higher influence on the estimate than those that are not.

2. Standard (homophilic) GNN models consist of a stack of graph convolution layers, each of which essentially multiplies the vertex features with the graph adjacency matrix, thereby propagating the features of a vertex to its neighbors [17, 34, 12]. For APPNP this hold especially true since the embedding vector of a vertex is defined as a weighted sum over the embeddings of its neighbors, where the weights are defined to be PPR probabilities (cf. Eq. (13)). Overall, ignoring the nonlinearities between convolutions, for GNN models, $k_{\mathrm{PPR}}(v, v')$ can be interpreted as an approximation of how similar the predictions of a GNN $h$ will be for two vertices $v, v'$.

Combining both points, the PPR kernel reweights vertices based on how similar the predictions of a GNN will be for them, i.e., SIS with PPR focuses on the region of the training data, where the predictive behavior of $h$ is similar to the test data. Empirically, we found that this works well for different combinations of models, datasets and types of distribution shifts. However, in principle, given additional knowledge, one could also design domain specific kernels.

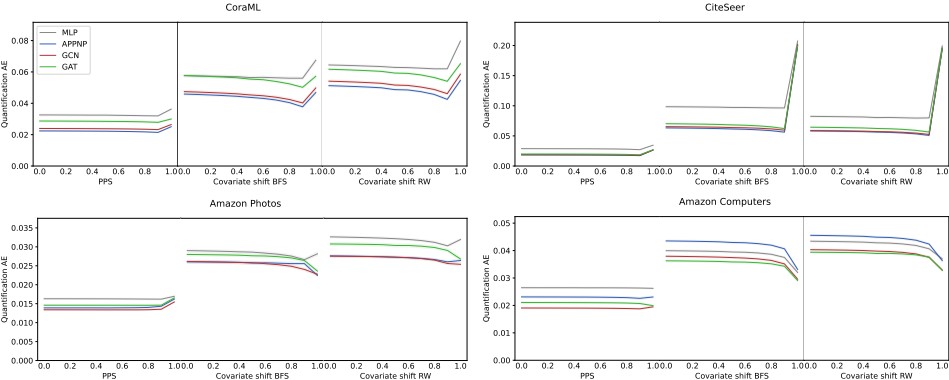

Figure 3: Quantification performance of SIS (with NACC) with the PPR kernel for different values of $\lambda$.

## B.2 Hyperparameter Evaluation of the PPR Kernel

As noted above, we use the interpolated PPR kernel from Eq. (15) with $\lambda = 0.9$ for all experiments. Intuitively, $\lambda$ controls the influence of training vertices $\mathcal{D}_L$ that are far-away from the test vertices $\mathcal{V}_U$. If $\lambda = 1$, far-away vertices are effectively ignored. If $\lambda < 1$, all vertices are considered at least to some degree, but the influence of far-away vertices is reduced. A large $\lambda$ can have the advantage of reducing the influence of irrelevant or misleading vertices from different regions of the graph. However, if too many vertices are excluded, the effective sample size for the confusion matrix estimate is reduced, making it more noisy, which can, in turn, degrade performance.

Figure 3 shows the quantification performance of SIS with the PPR kernel for different values of $\lambda$. For CoraML and CiteSeer, $\lambda < 1$ clearly outperforms $\lambda = 1$, with $\lambda \approx 0.9$ performing very well. For the Amazon Photos and Computers datasets, $\lambda = 1$ performs best. Interestingly, for CoraML and CiteSeer, the quantification performance for $\lambda = 1$ is significantly worse than for $\lambda < 1$, while for the Amazon datasets, $\lambda = 1$ performs better than $\lambda < 1$. The reason for this discrepancy is that the CoraML and CiteSeer graphs contain multiple small components that are disconnected from the main connected component; for structurally shifted test distributions that lie within one of those small components, the PPR-based confusion matrix estimates are then based on very few vertices, making it very noisy and thereby degrading performance. By assigning at least a small weight to all vertices, e.g., via $\lambda = 0.9$, the effective sample size is increased, which leads to better estimates on those datasets. For this reason, we found that a value of $\lambda$ that is slightly smaller than 1 works well for most datasets, which is why we chose $\lambda = 0.9$ as a default.

## B.3 Evaluation of the *Shortest path* Kernel

For comparison, we also evaluated an alternative to the PPR kernel. This alternative kernel is based on the *shortest path* (SP) distance between vertices instead of RW probabilities. We define this shortest-path kernel as

$$k_{\mathrm{SP}}(v, v') = \exp(-\gamma \cdot d_{\mathrm{SP}}(v, v')) , \tag{16}$$

where $d_{\mathrm{SP}}(v, v')$ is the length of the shortest path length between $v$ and $v'$ and $\gamma > 0$ a tunable hyperparameter.

Figure 4 shows the quantification performance of SIS with the shortest-path kernel for different values of $\gamma$. Under PPS, increasing $\gamma$ leads to a decrease in performance on all datasets except CiteSeer. Under covariate shift (both, BFS and RW), increasing $\gamma$ generally improves the quantification performance.

This is plausible, since under PPS, all training vertices are equally important, while under covariate shift, the training vertices that are close to the test vertices are more important than those that are far away. By increasing $\gamma$, the kernel becomes more peaked around close vertices. Under covariate shift, where far-away vertices are less important, this leads to better quantification performance, while under PPS more aggressive reweighting decreases performance.

Table 3: Comparison of SIS quantification with the SP and the PPR kernel.

| Model & Shift | Quantifier | CoraML AE | RAE | CiteSeer AE | RAE | A. Photos AE | RAE | A. Computers AE | RAE | PubMed AE | RAE | Avg. Rank AE | RAE |
|---|---|---|---|---|---|---|---|---|---|---|---|---|---|
| PPS | MLPE | .0903 | .5692 | .0469 | .2623 | .0924 | .6660 | .0770 | .5358 | .1268 | .3948 | - | - |
| MLP PPS | PCC | .0827 | .8565 | .0361 | .2782 | .0497 | 1.105 | .0533 | .6342 | .0470 | .1870 | 5.6 | 5.8 |
| | PACC | .0481 | .4186 | .0336 | .2271 | .0191 | **.3036** | .0334 | .3690 | .0181 | .0649 | 3.2 | 3.0 |
| | NEIGH PACC | .0326 | .2865 | .0288 | .1908 | .0163 | .3595 | .0265 | .2936 | .0187 | .0649 | 2.2 | 2.6 |
| | SP PACC | .0656 | .5871 | .0573 | .3916 | .0235 | .3243 | .0406 | .4278 | .0296 | .1129 | 5.0 | 4.4 |
| | SP NEIGH PACC | .0421 | .4114 | .0400 | .2702 | .0194 | .3575 | .0283 | .3178 | .0327 | .1253 | 4.0 | 3.8 |
| | PPR NEIGH PACC | **.0320** | **.2847** | **.0271** | **.1799** | **.0162** | .3345 | **.0263** | **.2913** | **.0178** | **.0616** | **1.0** | **1.4** |
| GAT PPS | PCC | .0479 | .5323 | .0219 | .1573 | .0314 | .9570 | .0398 | .4674 | .0463 | .1911 | 5.6 | 5.6 |
| | PACC | .0297 | .2660 | .0192 | .1262 | .0147 | **.2776** | .0217 | .2326 | **.0176** | **.0635** | 2.6 | 2.0 |
| | NEIGH PACC | .0287 | .2438 | .0199 | .1307 | **.0146** | .3249 | .0211 | .2271 | .0192 | .0694 | 2.4 | 2.4 |
| | SP PACC | .0375 | .3481 | .0332 | .2192 | .0202 | .4594 | .0239 | .2683 | .0309 | .1246 | 5.0 | 4.8 |
| | SP NEIGH PACC | .0369 | .3252 | .0321 | .2131 | .0195 | .5628 | **.0207** | .2426 | .0341 | .1378 | 4.0 | 4.6 |
| | PPR NEIGH PACC | **.0279** | **.2414** | **.0187** | **.1235** | **.0146** | .3466 | **.0207** | **.2238** | .0181 | .0655 | 1.4 | 1.6 |
| GCN PPS | PCC | .0438 | .4697 | .0221 | .1574 | .0315 | .8508 | .0391 | .4667 | .0405 | .1665 | 5.6 | 5.6 |
| | PACC | .0246 | .2216 | .0190 | .1259 | **.0122** | **.2056** | .0228 | .2411 | **.0161** | **.0591** | 2.4 | 2.2 |
| | NEIGH PACC | .0239 | **.2073** | .0188 | .1253 | .0134 | .2920 | .0191 | .2054 | .0181 | .0659 | 2.2 | 2.0 |
| | SP PACC | .0315 | .3089 | .0302 | .2045 | .0182 | .3994 | .0279 | .3051 | .0298 | .1276 | 5.0 | 4.8 |
| | SP NEIGH PACC | .0308 | .2862 | .0301 | .2013 | .0181 | .5700 | .0223 | .2436 | .0319 | .1386 | 4.2 | 4.6 |
| | PPR NEIGH PACC | **.0232** | **.2073** | **.0176** | **.1177** | .0135 | .3329 | **.0188** | **.2029** | .0168 | .0613 | 1.6 | **1.8** |
| APPNP PPS | PCC | .0374 | .4124 | .0214 | .1509 | .0318 | .9795 | .0390 | .4657 | .0398 | .1664 | 5.6 | 5.6 |
| | PACC | .0217 | .1986 | .0184 | .1211 | **.0124** | **.2442** | .0256 | .2638 | **.0165** | **.0597** | **1.8** | 2.0 |
| | NEIGH PACC | .0224 | .1943 | .0184 | .1222 | .0139 | .3133 | .0231 | .2471 | .0187 | .0676 | 2.6 | 2.4 |
| | SP PACC | .0286 | .2933 | .0293 | .1969 | .0181 | .6573 | .0372 | .3970 | .0302 | .1284 | 4.4 | 4.6 |
| | SP NEIGH PACC | .0292 | .2723 | .0300 | .1993 | .0192 | .7014 | .0262 | .2859 | .0332 | .1444 | 5.0 | 4.8 |
| | PPR NEIGH PACC | **.0214** | **.1939** | **.0172** | **.1149** | .0143 | .3780 | **.0226** | **.2424** | .0175 | .0642 | 1.6 | **1.6** |
| BFS | MLPE | .1839 | 8.185 | .2676 | 15.22 | .1622 | 9.313 | .1180 | 5.666 | .3034 | 5.067 | - | - |
| MLP BFS | PCC | .1243 | 7.212 | .1588 | 14.84 | .0668 | 4.028 | .0662 | 3.635 | .0800 | 10.44 | 5.0 | 5.4 |
| | PACC | .0645 | 3.508 | .1158 | 10.63 | **.0237** | **.9928** | .0392 | **1.608** | .0816 | 7.663 | 3.0 | 2.2 |
| | NEIGH PACC | .0577 | 3.008 | .0984 | 8.845 | .0290 | 1.237 | .0400 | 1.817 | .0878 | **6.787** | 3.8 | 2.6 |
| | SP PACC | .0787 | 4.803 | .2134 | 20.36 | .0272 | 1.477 | .0358 | 1.736 | .0783 | 12.26 | 3.6 | 5.0 |
| | SP NEIGH PACC | .0715 | 4.636 | .2080 | 19.92 | .0294 | 1.519 | **.0342** | 1.682 | **.0756** | 10.20 | 3.2 | 4.0 |
| | PPR NEIGH PACC | **.0560** | **2.972** | **.0964** | **8.699** | .0266 | 1.097 | .0375 | 1.694 | .0840 | 6.833 | 2.4 | **1.8** |
| GAT BFS | PCC | .0741 | 4.840 | .0820 | 7.349 | .0291 | 1.757 | .0455 | 2.415 | **.0650** | 9.922 | 4.6 | 5.4 |
| | PACC | .0561 | 2.533 | .0656 | **5.347** | .0243 | **.7255** | .0331 | **.9463** | .0930 | **6.906** | **2.6** | **1.4** |
| | NEIGH PACC | .0577 | 2.548 | .0702 | 5.886 | .0280 | .8623 | .0363 | 1.234 | .1011 | 8.096 | 4.4 | 3.2 |
| | SP PACC | .0618 | 3.745 | .2058 | 19.24 | **.0232** | .9479 | **.0290** | 1.092 | .0784 | 9.279 | 3.0 | 4.0 |
| | SP NEIGH PACC | .0616 | 3.667 | .2030 | 19.24 | .0266 | 1.108 | .0319 | 1.343 | .0875 | 10.40 | 3.6 | 5.2 |
| | PPR NEIGH PACC | **.0502** | **2.179** | **.0621** | **5.134** | .0264 | .8485 | .0343 | 1.223 | .0962 | 7.892 | 2.8 | 1.8 |
| GCN BFS | PCC | .0539 | 3.489 | .0783 | 7.060 | .0256 | 1.513 | .0418 | 2.255 | **.0573** | 9.553 | 4.0 | 5.0 |
| | PACC | .0488 | 2.093 | .0637 | 5.267 | .0241 | **.5966** | .0401 | **.9320** | .0888 | **6.713** | 3.4 | **1.6** |
| | NEIGH PACC | .0474 | 2.020 | .0653 | 5.428 | .0261 | .6773 | .0379 | .9846 | .0977 | 7.994 | 4.2 | 2.4 |
| | SP PACC | .0561 | 3.503 | .2083 | 19.70 | **.0230** | .8628 | .0331 | 1.051 | .0785 | 10.40 | 3.4 | 5.0 |
| | SP NEIGH PACC | .0541 | 3.276 | .2053 | 19.45 | .0255 | 1.013 | **.0322** | 1.104 | .0855 | 11.08 | 3.6 | 5.0 |
| | PPR NEIGH PACC | **.0402** | **1.786** | **.0595** | **4.910** | .0240 | .7070 | .0351 | 1.004 | .0924 | 7.723 | **2.4** | 2.0 |
| APPNP BFS | PCC | .0469 | 3.074 | .0737 | 6.609 | .0271 | 1.492 | .0468 | 2.339 | **.0569** | 9.867 | 4.2 | 4.8 |
| | PACC | .0457 | 1.881 | .0603 | 4.944 | **.0225** | **.5731** | .0430 | **.9227** | .0927 | **7.449** | 2.8 | **1.4** |
| | NEIGH PACC | .0459 | 1.910 | .0633 | 5.243 | .0260 | .6585 | .0435 | .9606 | .1017 | 8.673 | 4.2 | 2.6 |
| | SP PACC | .0531 | 3.327 | .2081 | 19.54 | **.0223** | .8863 | **.0330** | 1.045 | .0799 | 10.94 | 3.2 | 5.0 |
| | SP NEIGH PACC | .0515 | 3.140 | .2037 | 19.29 | .0261 | 1.037 | .0344 | 1.148 | .0872 | 11.60 | 4.0 | 5.2 |
| | PPR NEIGH PACC | **.0378** | **1.615** | **.0562** | **4.600** | .0256 | .6993 | .0406 | .9747 | .0962 | 8.440 | 2.6 | 2.0 |
| RW | MLPE | .1832 | 6.430 | .2651 | 15.07 | .1594 | 8.278 | .1158 | 4.466 | .3025 | 3.073 | - | - |
| MLP RW | PCC | .1263 | 5.275 | .1494 | 13.84 | .0727 | 3.820 | .0718 | 3.224 | .0913 | 1.376 | 5.0 | 5.6 |
| | PACC | .0733 | 2.347 | .0869 | 7.425 | .0332 | 1.251 | .0471 | 1.837 | .0882 | .7452 | 3.0 | 2.6 |
| | NEIGH PACC | .0644 | 2.153 | .0824 | 7.105 | .0326 | 1.369 | .0434 | 1.859 | .0915 | .7066 | 2.6 | 2.4 |
| | SP PACC | .0923 | 3.841 | .2081 | 19.71 | .0384 | 1.805 | .0471 | 2.046 | .0926 | .8547 | 5.2 | 5.0 |
| | SP NEIGH PACC | .0798 | 3.580 | .2012 | 19.07 | .0340 | 1.670 | .0408 | 1.863 | .0949 | .8579 | 4.2 | 4.4 |
| | PPR NEIGH PACC | **.0620** | **2.040** | **.0798** | **6.894** | **.0303** | **1.201** | **.0406** | **1.735** | **.0880** | **.6960** | **1.0** | **1.0** |
| GAT RW | PCC | .0799 | 3.555 | .0766 | 6.693 | .0340 | 1.689 | .0500 | 2.195 | **.0691** | .7488 | 4.6 | 5.6 |
| | PACC | .0610 | 1.648 | .0594 | 4.563 | .0293 | **.7767** | .0382 | **.9639** | .0952 | .6084 | 2.8 | **1.6** |
| | NEIGH PACC | .0617 | 1.836 | .0644 | 5.184 | .0308 | .9098 | .0394 | 1.191 | .0977 | .6290 | 4.4 | 3.2 |
| | SP PACC | .0713 | 2.994 | .1995 | 18.61 | .0296 | 1.062 | **.0370** | 1.235 | .0908 | **.5995** | 3.0 | 3.6 |
| | SP NEIGH PACC | .0721 | 3.175 | .1996 | 18.79 | .0306 | 1.225 | .0380 | 1.421 | .0958 | .7424 | 4.6 | 5.2 |
| | PPR NEIGH PACC | **.0540** | **1.496** | **.0564** | **4.432** | **.0290** | .8826 | .0376 | 1.185 | .0951 | .6238 | 1.6 | 1.8 |
| GCN RW | PCC | .0539 | 2.085 | .0694 | 5.990 | .0276 | 1.247 | .0451 | 1.961 | **.0566** | **.4972** | 3.4 | 4.2 |
| | PACC | .0571 | 1.267 | .0554 | 4.204 | .0298 | **.6101** | .0428 | **.8952** | .0956 | .5915 | 4.4 | **1.8** |
| | NEIGH PACC | .0541 | 1.311 | .0588 | 4.637 | .0276 | .6622 | .0403 | .9491 | .0964 | .6260 | 3.8 | 3.0 |
| | SP PACC | .0646 | 2.515 | .1995 | 18.76 | .0291 | .8745 | .0388 | 1.048 | .0888 | .5576 | 4.2 | 4.2 |
| | SP NEIGH PACC | .0648 | 2.689 | .1971 | 18.63 | .0275 | 1.003 | **.0374** | 1.155 | .0933 | .7050 | 3.4 | 5.4 |
| | PPR NEIGH PACC | **.0461** | **1.034** | **.0524** | **4.049** | **.0256** | .6682 | .0375 | .9611 | .0934 | .6161 | **1.8** | 2.4 |
| APPNP RW | PCC | .0465 | 1.750 | .0659 | 5.638 | .0293 | 1.197 | .0504 | 2.016 | **.0546** | **.4160** | 3.8 | 4.2 |
| | PACC | .0527 | 1.121 | .0541 | 4.060 | .0282 | **.5726** | .0452 | **.8693** | .0979 | .5958 | 3.6 | **1.8** |
| | NEIGH PACC | .0512 | 1.229 | .0583 | 4.580 | .0276 | .6387 | .0456 | .9643 | .0999 | .6302 | 3.8 | 3.0 |
| | SP PACC | .0602 | 2.352 | .2005 | 18.83 | .0285 | .8829 | **.0398** | 1.089 | .0873 | .5390 | 3.8 | 4.2 |
| | SP NEIGH PACC | .0609 | 2.531 | .1972 | 18.66 | .0283 | 1.024 | .0402 | 1.237 | .0914 | .6818 | 4.0 | 5.4 |
| | PPR NEIGH PACC | **.0425** | **.8951** | **.0508** | **3.873** | **.0261** | .6387 | .0424 | .9700 | .0965 | .6157 | **2.0** | 2.4 |

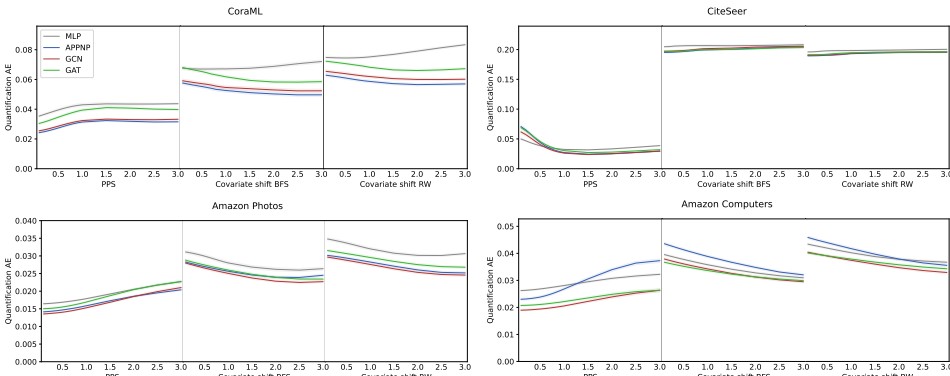

Figure 4: Quantification performance of SIS (with NACC) with the shortest-path kernel for different values of $\gamma$.

Table 3 compares the quantification performance of SIS with the SP and the PPR kernel. Since large $\gamma$ values generally lead to better performance under (structural) covariate shift, we used $\gamma = 3$ as a default for the reported SP kernel results. Overall, we find that the PPR kernel outperforms the SP kernel. Looking at the average ranks, we find that the SP kernel performs worst under PPS and best under BFS covariate shift. This is plausible, since the BFS-induced covariate shift samples vertices based on their distance to some root vertex; the test vertex density $q_V$ thus depends on SP distances. Nonetheless, the PPR kernel is still a better default choice.

## C  Runtime Analysis of SIS and NACC

Here, we analyze the runtime complexity of SIS for the PPR and SP kernels and provide additional information on their implementation. Additionally, we describe the complexity of NACC.

The main difference between standard ACC and ACC with SIS is the reweighting of training instances to account for covariate shift. Given a reweighted confusion matrix estimate, label prevalences are estimated using constrained optimization as described in Eq. (6), just like in standard ACC. The computational complexity of SIS thus is fully determined by the cost of computing $\rho_V(v)$ for all $v \in \mathcal{D}_L$ (see Eq. (11)). As defined in Eq. (12), $\rho_V(v)$ is estimated via KDE.

As explained in Section 4.1, we use the constant kernel $k_p = k_1$ to estimate $p_V$, since the training data is sampled uniformly at random in our experiments and only the test data is subject to distribution shift. The computational complexity of SIS is thus determined by the time it takes to compute the kernel values $k_q$. Let $T_k$ be the time it takes to compute the kernel values $\{k_q(v, v') \mid (v, y) \in \mathcal{D}_L, v' \in \mathcal{V}_U\}$. Based on those values, the time complexity of computing all weights $\rho(v)$ thus is $T = \mathcal{O}(T_k + |\mathcal{D}_L| \cdot |\mathcal{V}_U|)$. Here, $T_k$ depends on the choice of the kernel $k$.

### C.1  Complexity of the PPR Kernel

For the PPR kernel from Eq. (13), the random walk probabilities $\Pi_{v',v}^L$ have to be computed, where $L$ is the length of the random walk. The matrix $\Pi^L \in \mathbb{R}^{|\mathcal{V}| \times |\mathcal{V}|}$ is defined as

$$\Pi^L = \left( \alpha \mathbf{I} + (1-\alpha)\hat{\mathbf{A}} \right)^L,$$

where $\hat{\mathbf{A}} = \mathbf{D}^{-1}\mathbf{A}$ is the normalized (random-walk) adjacency matrix of the graph and $\alpha \in [0, 1]$ is the probability of not moving in a given step of the random walk. Naively, $\Pi^L$ can be computed via standard dense matrix multiplication in $\mathcal{O}(|\mathcal{V}|^{3 \log_2 L})$, or $\mathcal{O}(|\mathcal{V}|^{2.807 \log_2 L})$, using Strassen's algorithm. Since matrix multiplication is very parallelizable, this naive strategy can be sufficient, even for medium to large graphs. The experiments on the PubMed dataset, which is the largest dataset with synthetic shifts, with roughly 20k vertices, were conducted using dense matrix multiplication on the described hardware (see Appendix A).

If dense multiplication is not feasible and the adjacency matrix is sparse (which is typically the case for large real-world graphs), the PPR kernel can still be computed using sparse matrix multiplications. Since the sparsity of $\Pi^L$ decreases with increasing $L$, simply applying sparse matrix multiplication $L$ times is not feasible, as the resulting matrices quickly become dense. One way to tackle this "densification" problem has been proposed by Damke and Hüllermeier [4]. They propose to ensure a certain level of sparsity after each multiplication by pruning entries below a threshold $\delta$. This approximation allows for an efficient computation of $\Pi^L$ even for large graphs. We used this approach for the experiments with the "Twitch Gamers" dataset (over 168k vertices).

### C.2 Complexity of the Shortest path Kernel

For the SP kernel from Eq. (16), we need to compute the distances from the vertices in $\mathcal{D}_L$ to the vertices in $\mathcal{V}_U$. Given an undirected graph without edge weights, the distance from any node $v$ to all other nodes $v'$ can computed via BFS in $\mathcal{O}(|\mathcal{E}|)$, where $\mathcal{E}$ is the edge set of the graph. Since node distances are symmetric in undirected graphs, we can thus simply run BFS for each node in $\mathcal{D}_L$ or in $\mathcal{V}_U$. Starting the BFS traversals from the nodes in the smaller set, one then gets $T_k = \mathcal{O}(\min(|\mathcal{D}_L|, |\mathcal{V}_U|) \cdot |\mathcal{E}|)$.

On a more practical note, when one wants to quantify label prevalences for many test sets sampled from the same graph, it is best to compute the entire *all-pairs shortest path* distance matrix $D \in \mathbb{N}_0^{|\mathcal{V}| \times |\mathcal{V}|}$ in advance and to then use it as a lookup table for each quantification task. If the graph is sparse, i.e., $|\mathcal{E}| \ll |\mathcal{V}|^2$, $D$ can be computed fairly efficiently. Additionally, distance computation in graphs lends itself well to parallelization. For our implementation, we used the Python bindings of Nvidia's libcugraph library, which provides a fast implementation of BFS for GPUs[1].

### C.3 Complexity of NACC

The primary difference between standard ACC and *Neighborhood-aware ACC* (NACC) is the confusion matrix estimate $\hat{\mathbf{C}}$ and the distribution of predictions $\hat{Q}(\hat{Y})$ used in Eq. (6).

ACC uses a square $K \times K$ confusion matrix and a $K$-dimensional predictive distribution vector; both can be computed in $\mathcal{O}(|\mathcal{V}| + K^2)$ via a single pass over the vertices to count predicted label frequencies, followed by normalization. The objective of the constrained optimization problem from Eq. (6) can then be evaluated in $\mathcal{O}(K^2)$. The number of required objective evaluations depends on the used optimization algorithm; since the objective is quadratic, the number of optimizer evaluations can, however, be assumed to be in $\mathcal{O}(1)$ when using a (quasi-)Newtonian method. The number of parameters to the optimization problem is $K$, a quasi-Newtonian optimizer can thus compute and store estimates of the Hessian in $\mathcal{O}(K^2)$. Overall, the time complexity of ACC is thus $\mathcal{O}(|\mathcal{V}| + K^2)$.

NACC uses an overdetermined system of equations with a rectangular $K^2 \times K$ confusion matrix estimate and a $K^2$-dimensional predictive distribution vector. NACC computes the prediction frequencies of all class pairs $(j, k) \in \mathcal{Y}^2$, where the first class $j$ is the predicted label of a class and $k$ is the predicted majority label in its neighborhood. The majority label for all neighborhoods can be computed in $\mathcal{O}(|\mathcal{E}|)$. Combining this with the increased size of the confusion matrix, NACC thus has an overall time complexity of $\mathcal{O}(|\mathcal{V}| + |\mathcal{E}| + K^3)$. If the graph is sparse and the number of classes $K$ is relatively small, NACC scales well even to large graphs.

## D  Feature-based Importance Sampling for Graph Quantification

In Section 3.1, we introduced SIS as a method to account for structural covariate shift in graph quantification. SIS is based on the assumption that the marginal densities $p_V$ and $q_V$ of the training and test vertices can be estimated using vertex kernels $k_p$ and $k_q$ that capture structural similarity between vertices. While this is a reasonable assumption under structural covariate shift, one could, in principle, use other types of kernels to estimate vertex similarity. One natural alternative to the structural kernels (Eqs. (15) and (16)) are non-structural kernels that only consider vertex features. In homophilic graphs, the features of neighboring vertices tend to be similar, indicating that feature similarity could be used as a proxy for structural similarity.

---

[1] https://docs.rapids.ai/api/cugraph/legacy/api_docs/api/plc/pylibcugraph.bfs/

Table 4: Comparison of SIS with the PPR kernel and an inner product feature kernel.

| Model & Shift | Quantifier | CoraML AE | RAE | CiteSeer AE | RAE | A. Photos AE | RAE | A. Computers AE | RAE | PubMed AE | RAE | Avg. Rank AE | RAE |
|---|---|---|---|---|---|---|---|---|---|---|---|---|---|
| **MLP** PPS | PCC | .0827 | .8565 | .0361 | .2782 | .0497 | 1.105 | .0533 | .6342 | .0470 | .1870 | 7.0 | 7.0 |
| | PACC | .0481 | .4186 | .0336 | .2271 | .0191 | .3036 | .0334 | .3690 | **.0181** | .0649 | 4.4 | 4.2 |
| | NEIGH PACC | **.0326** | **.2865** | .0288 | .1908 | **.0163** | .3595 | **.0265** | **.2936** | .0187 | .0649 | 3.0 | 3.0 |
| | PPR PACC | .0486 | .4237 | .0327 | .2218 | .0192 | **.2881** | .0338 | .3708 | **.0179** | **.0641** | 4.8 | 3.6 |
| | PPR NEIGH PACC | **.0320** | **.2847** | **.0271** | **.1799** | .0162 | .3345 | **.0263** | **.2913** | **.0178** | **.0616** | 1.2 | 1.4 |
| | FEATURE PACC | .0487 | .4387 | .0326 | .2513 | .0191 | .3378 | .0326 | .3655 | **.0181** | .0658 | 4.6 | 5.2 |
| | FEATURE NEIGH PACC | .0344 | .3061 | .0297 | .2201 | **.0165** | .4188 | **.0260** | **.2939** | .0184 | **.0641** | 3.0 | 3.6 |
| **GCN** PPS | PCC | .0438 | .4697 | .0221 | .1574 | .0315 | .8508 | .0391 | .4667 | .0405 | .1665 | 7.0 | 7.0 |
| | PACC | .0246 | .2216 | .0190 | .1259 | **.0122** | **.2056** | .0228 | .2411 | .0161 | .0591 | 4.2 | 4.0 |
| | NEIGH PACC | .0239 | **.2073** | .0188 | .1253 | .0134 | .2920 | **.0191** | **.2054** | .0181 | .0659 | 4.2 | 3.6 |
| | PPR PACC | **.0234** | .2163 | **.0178** | **.1186** | **.0124** | .2295 | .0223 | .2386 | **.0151** | **.0555** | 2.4 | 3.0 |
| | PPR NEIGH PACC | **.0232** | **.2073** | **.0176** | **.1177** | .0135 | .3329 | **.0188** | **.2029** | .0168 | .0613 | 2.6 | 3.0 |
| | FEATURE PACC | **.0237** | .2176 | .0189 | .1312 | **.0123** | **.2103** | .0224 | .2374 | .0157 | **.0572** | 3.4 | 3.6 |
| | FEATURE NEIGH PACC | **.0236** | **.2064** | .0191 | .1328 | .0134 | .3049 | **.0189** | **.2038** | .0174 | .0629 | 4.2 | 3.8 |
| **APPNP** PPS | PCC | .0374 | .4124 | .0214 | .1509 | .0318 | .9795 | .0390 | .4657 | .0398 | .1664 | 7.0 | 7.0 |
| | PACC | .0217 | .1986 | .0184 | .1211 | **.0124** | **.2442** | .0256 | .2638 | .0165 | .0597 | 3.8 | 3.8 |
| | NEIGH PACC | .0224 | **.1943** | .0184 | .1222 | .0139 | .3133 | **.0231** | **.2471** | .0187 | .0676 | 4.8 | 4.2 |
| | PPR PACC | **.0203** | **.1926** | **.0171** | **.1132** | .0130 | .3058 | .0249 | .2566 | **.0154** | **.0558** | 2.0 | 2.0 |
| | PPR NEIGH PACC | .0214 | **.1939** | **.0172** | **.1149** | .0143 | .3780 | **.0226** | **.2424** | .0175 | .0642 | 3.2 | 3.2 |
| | FEATURE PACC | .0212 | **.1978** | .0183 | .1259 | **.0124** | **.2517** | .0255 | .2614 | **.0159** | **.0574** | 2.6 | 3.8 |
| | FEATURE NEIGH PACC | .0223 | **.1936** | .0188 | .1298 | .0140 | .3341 | **.0229** | **.2449** | .0182 | .0658 | 4.6 | 4.0 |
| **MLP** BFS | PCC | .1243 | 7.212 | .1588 | 14.84 | .0668 | 4.028 | .0662 | 3.635 | **.0800** | 10.44 | 6.2 | 7.0 |
| | PACC | .0645 | 3.508 | .1158 | 10.63 | .0237 | .9928 | .0392 | 1.608 | .0816 | 7.663 | 3.6 | 3.2 |
| | NEIGH PACC | **.0577** | **3.008** | **.0984** | **8.845** | .0290 | 1.237 | .0400 | 1.817 | .0878 | **6.787** | 4.6 | 3.0 |
| | PPR PACC | .0637 | 3.461 | .1162 | 10.74 | **.0222** | **.9079** | **.0370** | **1.509** | .0786 | 7.827 | 2.6 | 3.2 |
| | PPR NEIGH PACC | **.0560** | **2.972** | **.0964** | **8.699** | .0266 | 1.097 | **.0375** | 1.694 | .0840 | 6.833 | 2.6 | 2.4 |
| | FEATURE PACC | .0628 | 3.573 | .1563 | 14.82 | .0244 | 1.097 | .0397 | 1.761 | **.0780** | 7.685 | 3.6 | 4.8 |
| | FEATURE NEIGH PACC | **.0560** | 3.053 | .1409 | 13.39 | .0298 | 1.345 | .0399 | 1.937 | .0847 | **6.814** | 4.8 | 4.4 |
| **GCN** BFS | PCC | .0539 | 3.489 | .0783 | 7.060 | .0256 | 1.513 | .0418 | 2.255 | **.0573** | 9.553 | 5.0 | 6.6 |
| | PACC | .0488 | 2.093 | .0637 | 5.267 | .0241 | **.5966** | .0401 | **.9320** | .0888 | **6.713** | 4.6 | 2.8 |
| | NEIGH PACC | .0474 | 2.020 | .0653 | 5.428 | .0261 | .6773 | .0379 | .9846 | .0977 | 7.994 | 5.2 | 4.2 |
| | PPR PACC | **.0415** | **1.943** | **.0618** | **5.132** | **.0207** | **.5932** | **.0358** | **.9569** | .0840 | 6.727 | 1.8 | 2.2 |
| | PPR NEIGH PACC | **.0402** | **1.786** | **.0595** | **4.910** | .0240 | .7070 | **.0351** | 1.004 | .0924 | 7.723 | 2.0 | 3.6 |
| | FEATURE PACC | .0476 | 2.067 | .1020 | 9.301 | .0241 | **.5964** | .0400 | .9359 | .0870 | **6.723** | 4.4 | 3.4 |
| | FEATURE NEIGH PACC | .0460 | 1.990 | .1020 | 9.358 | .0259 | .6811 | .0377 | **.9909** | .0957 | 8.058 | 5.0 | 5.2 |
| **APPNP** BFS | PCC | .0469 | 3.074 | .0737 | 6.609 | .0271 | 1.492 | .0468 | 2.339 | **.0569** | 9.867 | 5.4 | 6.6 |
| | PACC | .0457 | 1.881 | .0603 | 4.944 | .0225 | **.5731** | .0430 | **.9227** | .0927 | **7.449** | 3.8 | 2.8 |
| | NEIGH PACC | .0459 | 1.910 | .0633 | 5.243 | .0260 | .6585 | .0435 | .9606 | .1017 | 8.673 | 5.8 | 4.6 |
| | PPR PACC | **.0380** | **1.729** | **.0574** | **4.705** | **.0213** | **.5823** | **.0395** | **.9331** | .0874 | 7.372 | 1.6 | 2.2 |
| | PPR NEIGH PACC | **.0378** | **1.615** | **.0562** | **4.600** | .0256 | .6993 | .0406 | .9747 | .0962 | 8.440 | 2.6 | 3.6 |
| | FEATURE PACC | .0443 | 1.852 | .0985 | 8.937 | **.0220** | **.5707** | .0425 | .9178 | .0909 | **7.475** | 3.4 | 2.8 |
| | FEATURE NEIGH PACC | .0444 | 1.869 | .0992 | 9.057 | .0257 | .6678 | .0430 | .9647 | .0997 | 8.750 | 5.4 | 5.4 |
| **MLP** RW | PCC | .1263 | 5.275 | .1494 | 13.84 | .0727 | 3.820 | .0718 | 3.224 | .0913 | 1.376 | 6.8 | 7.0 |
| | PACC | .0733 | 2.347 | .0869 | 7.425 | .0332 | 1.251 | .0471 | 1.837 | **.0882** | **.7452** | 4.2 | 3.4 |
| | NEIGH PACC | .0644 | 2.153 | **.0824** | **7.105** | .0326 | 1.369 | .0434 | 1.859 | .0915 | **.7066** | 3.4 | 3.2 |
| | PPR PACC | .0743 | 2.417 | .0899 | 7.766 | .0315 | **1.144** | .0450 | **1.765** | .0864 | **.7580** | 3.6 | 3.4 |
| | PPR NEIGH PACC | **.0620** | **2.040** | **.0798** | **6.894** | **.0303** | 1.201 | **.0406** | **1.735** | .0880 | **.6960** | 1.4 | 1.2 |
| | FEATURE PACC | .0740 | 2.445 | .1275 | 11.62 | .0341 | 1.341 | .0479 | 1.970 | **.0858** | **.7662** | 4.8 | 5.6 |
| | FEATURE NEIGH PACC | .0644 | 2.229 | .1202 | 10.98 | .0331 | 1.456 | .0434 | 1.956 | .0890 | **.7061** | 3.8 | 4.2 |
| **GCN** RW | PCC | .0539 | 2.085 | .0694 | 5.990 | .0276 | 1.247 | .0451 | 1.961 | **.0566** | **.4972** | 4.4 | 5.4 |
| | PACC | .0571 | 1.267 | .0554 | **4.204** | .0298 | **.6101** | .0428 | **.8952** | .0956 | .5915 | 5.6 | 3.0 |
| | NEIGH PACC | .0541 | 1.311 | .0588 | 4.637 | .0276 | **.6622** | .0403 | .9491 | .0964 | .6260 | 4.8 | 5.0 |
| | PPR PACC | .0494 | **1.089** | .0527 | 4.001 | .0266 | **.5993** | .0392 | **.9029** | .0925 | .5778 | 2.0 | **1.8** |
| | PPR NEIGH PACC | **.0461** | **1.034** | .0524 | 4.049 | .0256 | **.6682** | **.0375** | .9611 | .0934 | .6161 | 1.4 | 4.0 |
| | FEATURE PACC | .0559 | 1.234 | .0767 | 6.490 | .0298 | **.6110** | .0426 | **.8962** | .0944 | .5873 | 5.6 | 3.4 |
| | FEATURE NEIGH PACC | .0526 | 1.267 | .0796 | 6.913 | .0273 | **.6644** | .0401 | .9552 | .0948 | .6226 | 4.2 | 5.4 |
| **APPNP** RW | PCC | .0465 | 1.750 | .0659 | 5.638 | .0293 | 1.197 | .0504 | 2.016 | **.0546** | **.4160** | 4.6 | 5.4 |
| | PACC | .0527 | 1.121 | .0541 | 4.060 | .0282 | **.5726** | .0452 | **.8693** | .0979 | .5958 | 5.2 | 3.2 |
| | NEIGH PACC | .0512 | 1.229 | .0583 | 4.580 | .0276 | .6387 | .0456 | .9643 | .0999 | .6302 | 5.2 | 5.0 |
| | PPR PACC | .0448 | **.9489** | **.0501** | **3.735** | .0255 | **.5650** | .0418 | **.8731** | .0944 | .5821 | 1.4 | 1.8 |
| | PPR NEIGH PACC | **.0425** | **.8951** | .0508 | 3.873 | .0261 | .6387 | **.0424** | .9700 | .0965 | .6157 | 2.0 | 3.8 |
| | FEATURE PACC | .0515 | 1.090 | .0744 | 6.223 | .0279 | **.5711** | .0450 | **.8672** | .0968 | .5911 | 4.8 | 3.0 |
| | FEATURE NEIGH PACC | .0498 | 1.189 | .0776 | 6.666 | .0273 | .6427 | .0452 | .9674 | .0983 | .6241 | 4.8 | 5.8 |

One advantage of a feature-based kernel is that it can be applied even in the absence of a graph structure, i.e., when only vertex features are available. Second, even if the graph structure is available, feature-based kernels can be more efficient to compute than structural kernels, especially on large graphs. To evaluate the suitability of feature-based kernels for graph quantification under structural covariate shift, we implemented a feature-based version of SIS using the inner product between vertex features as a kernel. Table 4 compares the quantification performance of this feature-based SIS variant with with SIS+PPR. Overall, we find that the feature-based SIS performs significantly worse than SIS with the PPR kernel across models and datasets. This demonstrates that structure-based vertex kernels are better suited to account for structural covariate shift in graph quantification than a purely feature-based kernel.

