# OpenReview forum: "Adjusted Count Quantification Learning on Graphs"
_NeurIPS.cc/2025/Conference — NeurIPS 2025 poster_

### Official Review · Reviewer_ZwZ8 · 2025-07-02

**Clarity:** 2
**Significance:** 2
**Originality:** 2
**Rating:** 3
**Confidence:** 3

**Summary:**

This paper tackles the problem of quantification learning (QL), that is, estimating label distributions instead of individual labels using the  graph-structured data. This problem is not explored for graph data yet. The authors identify limitations of existing approaches, particularly the adjusted classify & count (ACC) method under prior probability shift assumptions and this usually fails in graphs due to structural covariate shifts. Accordingly, the authors propose two key innovations:
(a) Structural Importance Sampling (SIS), which generalizes ACC to handle covariate shifts using kernel density estimation, and
(b) Neighborhood-aware ACC (NACC), which leverages neighborhood structure to improve class distinguishability in non-homophilic graphs.
The authors also conducted thorough experiments using 5 graph datasets and 3 types of distribution shift (PPS, BFS-based, and RW-based covariate shift) and demonstrated that their approaches outperform the existing baselines (measured using 2 metrics: absolute error (AE) and relative absolute error (RAE)).

**Questions:**

Please answer the following questions:

(a) SIS requires an estimate of the sampling distribution, which in practice assumes knowledge of the structural data generation or sampling process. This is a non-trivial requirement in real-world applications. How to address this challenge? With this, the potential impact of the proposed approach becomes limited. What are the alternatives to avoid this? Any thoughts?

(b) SIS depends on graph kernels and density estimation. They are typically compute heavy tasks and may not scale to large graphs. How to address this computational scaling issue?

(c) The authors work with synthetically induced distribution shifts on standard node classification datasets. These artificial or generated distribution shifts might not capture realistic scenarios as they are more complex in practice. How would you tackle this limitation in your experiments?

**Ethical Concerns:**

["NO or VERY MINOR ethics concerns only"]

**Limitations:**

Below are my key comments and concerns on this paper:

(a) SIS requires an estimate of the sampling distribution, which in practice assumes knowledge of the structural data generation or sampling process. This is a non-trivial requirement in real-world applications. How to address this challenge? With this, the potential impact of the proposed approach becomes limited. What are the alternatives to avoid this? Any thoughts?

(b) SIS depends on graph kernels and density estimation. They are typically compute heavy tasks and may not scale to large graphs. How to address this computational scaling issue?

(c) The authors work with synthetically induced distribution shifts on standard node classification datasets. These artificial or generated distribution shifts might not capture realistic scenarios as they are more complex in practice. How would you tackle this limitation in your experiments?

**Quality:**

2

**Strengths And Weaknesses:**

Below are my key comments and concerns on this paper:

(a) SIS requires an estimate of the sampling distribution, which in practice assumes knowledge of the structural data generation or sampling process. This is a non-trivial requirement in real-world applications. How to address this challenge? With this, the potential impact of the proposed approach becomes limited. What are the alternatives to avoid this? Any thoughts?

(b) SIS depends on graph kernels and density estimation. They are typically compute heavy tasks and may not scale to large graphs. How to address this computational scaling issue?

(c) The authors work with synthetically induced distribution shifts on standard node classification datasets. These artificial or generated distribution shifts might not capture realistic scenarios as they are more complex in practice. How would you tackle this limitation in your experiments?

---

> ### Author Rebuttal · Authors · 2025-07-31
>
> We appreciate the constructive feedback by the reviewer and will address the raised points individually.
>
> ### (a) Kernel-selection
>
> We fully agree that the relation between the data generating process and the kernel selection deserves further attention.
> To understand this relation, we will proceed in two steps:
> 1. What can and should one know about the data generating process?
> 2. How close does the kernel have to be aligned with the data generating process?
>
> Ultimately, the difficulty of determining whether a kernel is appropriate for a given problem depends on the amount of information available about the instance sampling process, i.e., about $P(V)$ and $Q(V)$.
> In practice, it is not unrealistic to assume that at least some information about those distributions is given.
>
> For example, consider a setting where the test nodes represent the social circle of a randomly selected person from a given community.
> Such test sets are natural if, for example, one wants to estimate the homogeneity of opinions among the friends (or friends of friends) of a person.
> If we know that the test sets consist of such connected social circles, this knowledge can be used to design a kernel, i.e., to estimate how likely it is that a given labeled vertex $v$ from $\mathcal{D}_L$ is sampled via the test distribution $Q(V)$, given samples $\mathcal{V}_U$.
> For this example, the probability of sampling $v$ corresponds to the probability of $v$ being part of the given social circle, which, in turn, depends on the distance between $v$ and the samples $\mathcal{V}_U$.
> A shortest-path kernel, as described in Eq. (S2) of the supplement, or some variant of it, is a reasonable choice in this case, as it determines the likelihood of a sample based on the distance between vertices.
> The parameterization of such a distance-based kernel then depends on properties of the social network, e.g., average node degrees and edge homophily.
> The kernel parameters should be treated as hyperparameters that need empirical tuning.
>
> While it is unrealistic to analytically derive an "optimal" kernel for real-world quantification problems, we hope that the above example illustrates that $k$ can be chosen in an informed manner without relying on pure trial-and-error.
>
> However, even if we have incomplete or incorrect information about the data generating process, SIS can be effective.
> Note that the effectiveness of SIS does not strictly depend on the accuracy of the density estimate $\hat{q}$; more precisely, the confusion matrix estimate $\hat{C}$ can be accurate, even if $\hat{q}$ deviates from $q$.
> Ultimately, we are only interested in whether the predictive quality of the classifier $h$ on $\hat{q}$ is similar to the predictive quality of $h$ on $\hat{q}$.
> From this perspective, the PPR kernel can then be seen as a good "default" choice because it (approximately) describes the GNN prediction similarity between vertices.
> See Section B of the supplementary PDF for further discussion of this.
>
> The general appropriateness of the PPR kernel is supported empirically by Table S1 of the supplement.
> There, we compare PPR against an alternative shortest-path-based kernel, which, at first glance, might appear to be a better match than PPR for the BFS-based covariate shift.
> Despite the apparent mismatch between PPR and BFS sampling, the PPR kernel outperforms the shortest-path kernel.
>
> Last, an evaluation of the influence of the hyperparameters of the shortest-path kernel is provided in Section B.3 of the supplement.
>
> ### (b) Computational complexity
>
> An analysis of the runtime of SIS and NACC was omitted from the main paper due to the page limit.
> We discuss the time complexity and scalability of our approach in Section C of the appendix.
>
> Naively, the PPR densities can be computed in $O(|V|^{3 \log_2 L})$; this does not scale to large graphs.
>
> To apply SIS with the PPR kernel to large graphs anyway, there are multiple options:
> 1. The kernel is not evaluated for all vertex pairs. Thus, computing a power of the entire adjacency matrix is not necessary. Instead, only train-test-pairs have to be computed. If either the training or the test data is small, one can compute the relevant kernel values in $O((|V|^2 \cdot \min ( |D_L|, |\mathcal{V}_U|))^{\log_2 L})$.
> 2. For densely connected graph, matrix multiplication can be sped up using Strassen's algorithm.
> 3. For sparse graphs, performance can be further improved via sparse-sparse or sparse-dense matrix multiplications. Combined with option 1, the relevant part of the PPR kernel can be computed in $O((|\mathcal{E}| \cdot \min ( |D_L|, |\mathcal{V}_U|))^{\log_2 L})$ via sparse-dense matrix multiplications.
>
> As described in (c), we have conducted an additional experiment on the large-scale Twitch Gamers dataset, containing ~168k vertices.
> Using the techniques described above (partial kernel eval, sparse matrix multiply), computing the PPR kernel is feasible even for large graphs.
>
> ### (c) Synthetic evaluation only
>
> Since existing node-classification benchmark datasets generally do not provide enough information to extract "natural" structural shifts, a more realistic evaluation is not trivial; for this reason we decided to induce three different types of distribution shifts synthetically on real datasets.
> To further strengthen our claim that SIS is effective at tackling structural covariate shift, we have created a more realistic graph quantification task.
>
> Social network datasets are a promising source of realistic structural covariate shifts.
> The "Twitch Gamers" dataset [1] is one such dataset; it consists of ~168k vertices and ~6.8 million edges where vertices represent Twitch accounts and edges represent followership relations.
> Each vertex is annotated with the language of the corresponding user, whether the user streams explicit content and a number of other features.
>
> To provide further insights into the effectiveness of SIS under "real" covariate shift, we conducted additional experiments using the "Twitch Gamers" dataset.
> We induced covariate shift by using a random subset of all users as training data and then selected different random subsets of the remaining users as test data where each test sample only contains users speaking a single language (EN, DE, FR, RU).
> Sampling the test data based on language induces a natural structural covariate shift, since users tend to follow users speaking the same language.
> We use the binary "explicit content" feature of the users as the target.
> The following table shows the AE achieved by different quantifiers on different language-filtered test splits.
> All quantifiers use an APPNP node classifier as their base model.
>
> |              |        EN |        DE |        FR |        RU |
> | ------------ | --------: | --------: | --------: | --------: |
> | CC           | **.0040** |     .1187 |     .1147 |     .1771 |
> | ACC          |     .0208 |     .0108 | **.0888** |     .0887 |
> | SIS-PPR ACC  |     .0107 | **.0076** |     .0899 | **.0801** |
> | SIS-PPR NACC |     .0958 |     .3991 |     .1768 |    .2553 |
>
> Since EN is the majority language in the dataset, there is barely any distribution shift between training and the EN test set; therefore, the unadjusted CC approach performs best.
> For all other languages, quantification performance is significantly improved via ACC.
> SIS with the PPR kernel further improves performance for the DE and RU test sets; on the FR test set, standard ACC performs slightly better than SIS.
> Adding NACC increases the quantification error; class identifiability is not really an issue in binary quantification.
> Overall, this experiment indicates that SIS can also deal with real-world structural covariate shift.
>
> We would like to thank the reviewer once again for their constructive critique of our work! We hope that we were able to answer your questions and clear up any concerns.
>
> ---
> 1. Rozemberczki, B., Sarkar, R.: Twitch Gamers: a Dataset for Evaluating Proximity Preserving and Structural Role-based Node Embeddings (2021).

---

> > ### Comment · Area_Chair_5Bqz · 2025-08-04
> >
> > Dear reviewer,
> >
> > Please engage in the discussion with the authors. The discussion period will end in a few days.

---

> > > ### Comment · Area_Chair_5Bqz · 2025-08-06
> > >
> > > Dear Reviewer,
> > >
> > > This is a reminder that we are three days away from the end of the discussion phase. Please engage with the authors as soon as possible to ensure they have sufficient time to respond.
> > >
> > > Note that not participating in the discussion may be considered a breach of the NeurIPS Code of Conduct.
> > >
> > > Best regards,
> > >
> > > The Area Chair

---

> > ### Author Response · Authors · 2025-08-08
> >
> > Dear reviewer,
> >
> > We hope that we were able to address your concerns in our rebuttal. If you have any further questions, we are happy to answer them in the time remaining.

---

### Official Review · Reviewer_ohSm · 2025-07-03

**Clarity:** 2
**Significance:** 1
**Originality:** 2
**Rating:** 1
**Confidence:** 3

**Summary:**

This paper studies the quantification learning problem in the context of graph-structured data. The quantification learning aims to predict the label distribution of a set of instances. Compared to previous clustering based models, this paper extends the popular Adjusted Classify & Count (ACC) method to graphs, and proposes structural importance sampling (SIS) for covariate shift and Neighborhood-aware ACC to improve the quantification in the presence of non-homophilic edges.

**Questions:**

see above

**Ethical Concerns:**

["NO or VERY MINOR ethics concerns only"]

**Limitations:**

see above

**Quality:**

1

**Strengths And Weaknesses:**

The strengths are as follows:
1.Clear problem demonstration. The definition of quantification learning on graphs is well demonstrated, including its necessity, potential application, difference from traditional instance classification on graphs.
2.Good background writing. The types of distribution shift, like concept shift, covarite shift and prior probability shift are well illustrated.

The weakness are as follows:
1.Although the quantification learning sounds a reasonable research prolem on graphs, I still think the instanece classification on graphs, together with aggregation models can well handle this problem. To be specific, the instance classification on graphs is well studied and many works can reach high performance on the used datasets, Citeseer, Pubmed and so on. Also, the results in Table 1 with AE and RAE seem not good metrics for researchers to understand how good the performance is. Any other metrics like classification acc?
2.The used datasets are limited. Any true benchmark datasets for this quantification task? Not the generated datasets from common GNN datasets.
3.The experiments are simple and monotonous, which is not convincing to illustrate the model's effectiveness.

---

> ### Author Rebuttal · Authors · 2025-07-31
>
> Thank you for your feedback and for sharing your concerns! We will address them in order.
>
> ### 1. Why quantification? Why AE and RAE?
>
> > I still think the instanece classification on graphs, together with aggregation models can well handle this problem
>
> As described in the introduction, while a perfect classifier also solves the quantification problem perfectly, a classifier that is "just" good, but not perfect, can be a poor quantifier and vice versa (a poor classifier can be a good quantifier).
> See Esuli et al. [1, Section 1.2] for a more in-depth explanation of why classification and quantification are two distinct tasks.
>
> In practice, vertex classifiers are rarely perfect. Consider Table 1, the *PCC* rows show the quantification performance of a naive classification-based quantification without any adjustments for different combinations of GNN-based classifiers and datasets.
> The *PACC* rows show the quantification performance of the classifiers with quantification-specific adjustments.
>
> If we compare PCC and PACC, PACC consistently and significantly outperforms PCC across classifier-types and datasets (both, in terms of absolute error and relative absolute error).
> The absolute error of PCC is oftentimes more than twice as large as that of the quantification-specific PACC approach.
> Even for the APPNP classifier, which is the most accurate among the evaluated ones, PACC outperforms PCC.
>
> Overall, this is very strong empirical evidence for usefulness of quantification in the graph setting.
> From this perspective, standard classification approaches cannot handle this problem well.
>
> > AE and RAE seem not good metrics for researchers to understand how good the performance is. Any other metrics like classification acc?
>
> In the literature on quantification, AE and RAE are among the most commonly used metrics to evaluate quantification performance. See Esuli et al. [1, Section 3.1] for an overview.
>
> To avoid misunderstandings, note that quantification and classification are two distinct tasks.
> In quantification, our goal is to predict a class distribution for a given dataset; in classification we want to predict a class for each instance in a given dataset.
> Thus, quantification can be seen as a dataset-level task, while classification is an instance-level task.
>
> For example, in binary quantification we might be interested in predicting the percentage of a population with a given disease (e.g., 23% have the disease, 77% do not).
> If our quantifier now predicts that only 21% of the population has the disease (79% do not), AE measures the quantification error as $\frac{1}{2}(|0.21 - 0.23| + |0.79 - 0.77|) = 0.02$.
>
> Classification metrics, such as accuracy, cannot be used to evaluate quantification performance.
>
> ### 2. More realistic evaluation
>
> Since existing node-classification benchmark datasets generally do not provide enough information to extract "natural" structural shifts, a more realistic evaluation is not trivial; for this reason we decided to induce three different types of distribution shifts synthetically on real datasets.
> To further strengthen our claim that SIS is effective at tackling structural covariate shift, we have created a more realistic graph quantification task.
>
> Social network datasets are a promising source of realistic structural covariate shifts.
> The "Twitch Gamers" dataset [2] is one such dataset; it consists of ~168k vertices and ~6.8 million edges where vertices represent Twitch accounts and edges represent followership relations.
> Each vertex is annotated with the language of the corresponding user, whether the user streams explicit content and a number of other features.
>
> To provide further insights into the effectiveness of SIS under "real" covariate shift, we have conducted additional experiments using the "Twitch Gamers" dataset.
> We induced covariate shift by using a random subset of all users as training data and then selected different random subsets of the remaining users as test data where each test sample only contains users speaking a single language (EN, DE, FR, RU).
> Sampling the test data based on language induces a natural structural covariate shift, since users tend to follow users speaking the same language.
> We use the binary "explicit content" feature of the users as the target.
> The following table shows the AE achieved by different quantifiers on different language-filtered test splits.
> All quantifiers use an APPNP node classifier as their base model.
>
> |              |        EN |        DE |        FR |        RU |
> | ------------ | --------: | --------: | --------: | --------: |
> | CC           | **.0040** |     .1187 |     .1147 |     .1771 |
> | ACC          |     .0208 |     .0108 | **.0888** |     .0887 |
> | SIS-PPR ACC  |     .0107 | **.0076** |     .0899 | **.0801** |
> | SIS-PPR NACC |     .0958 |     .3991 |     .1768 |    .2553 |
>
> Since EN is the majority language in the dataset, there is barely any distribution shift between training and the EN test set; therefore, the unadjusted CC approach performs best.
> For all other languages, quantification performance is significantly improved via ACC.
> SIS with the PPR kernel further improves performance for the DE and RU test sets; on the FR test set, standard ACC performs slightly better than SIS.
> Adding NACC increases the quantification error; class identifiability is not really an issue in binary quantification.
> Overall, this experiment indicates that SIS can also deal with real-world structural covariate shift.
>
> ### 3. Monotonous experiments
>
> >The experiments are simple and monotonous, which is not convincing to illustrate the model's effectiveness.
>
> We appreciate the feedback!
> For space reasons, we found a large table to be most space-efficient to represent our empirical results.
> While the presentation might be visually monotonous, our experiments do cover three different types of distribution shift across all combinations of 5 classifiers, 5 datasets and 5 quantification approaches.
>
> To quickly get an overall idea of the effectiveness of SIS and NACC, we included the column *Avg. Rank* which ranks the different quantification approaches for each shift setting and classifier from best (typeset in **bold**) to worst.
> Overall, either our proposed *SIS PACC* or our proposed *SIS NEIGH PACC* (SIS combined with NACC) achieves the best average quantification performance in all settings, showing that they are, indeed, effective.
>
> To make the date more digestible, we could split up the three shift settings into separate tables and, if the space permits, add some plots to make general trends in the data more visually apparent.
>
> Last, we want to note that you can find additional experimental evaluations in the *supplementary PDF*. There, we included plots evaluating the effect of kernel hyperparameters on quantification performance, along with a more in-depth analysis of the results.
>
>
> We hope that we were able to address your concerns and that you find these additional explanations helpful.
>
>
> ---
> 1. Esuli, A. et al.: Learning to Quantify. Springer, Cham (2023).
> 2. Rozemberczki, B., Sarkar, R.: Twitch Gamers: a Dataset for Evaluating Proximity Preserving and Structural Role-based Node Embeddings (2021).

---

> > ### Comment · Area_Chair_5Bqz · 2025-08-04
> >
> > Dear reviewer,
> >
> > Please engage in the discussion with the authors. The discussion period will end in a few days.

---

> > > ### Comment · Area_Chair_5Bqz · 2025-08-06
> > > **please engage**
> > >
> > > Dear Reviewer,
> > >
> > > This is a reminder that we are three days away from the end of the discussion phase. Please engage with the authors as soon as possible to ensure they have sufficient time to respond.
> > >
> > > Note that not participating in the discussion may be considered a breach of the NeurIPS Code of Conduct.
> > >
> > > Best regards,
> > >
> > > The Area Chair

---

### Official Review · Reviewer_GwGf · 2025-07-03

**Clarity:** 3
**Significance:** 3
**Originality:** 3
**Rating:** 5
**Confidence:** 2

**Summary:**

The paper tackles the problem of estimating label prevalences on graph-structured data (graph quantification learning) where prior work was confined to simple node-clustering heuristics. It extends the Adjusted Classify & Count (ACC) family beyond the prior probability shift regime by introducing **Structural Importance Sampling (SIS)**, a covariate-shift-aware reweighting scheme that builds a kernelised confusion matrix using personalised PageRank densities so each training vertex votes in proportion to how likely it would have been sampled in the test graph. To resolve the class-identifiability failures that arise when non-homophilic edges blur decision boundaries, the authors add **Neighborhood-aware ACC (NACC)**: every prediction is paired with the majority label of its 1-hop neighbourhood, enlarging the observable confusion signature without retraining the base classifier. Both upgrades drop straight into the standard constrained least-squares ACC solver and gracefully collapse back to vanilla ACC when the shift is pure PPS.

Empirically, SIS and NACC are plugged on top of four backbones (MLP, GCN, GAT, APPNP) and evaluated on five benchmark graphs (CoraML, CiteSeer, PubMed, Amazon-Photos, Amazon-Computers) under three shift scenarios - prior-probability, BFS-induced covariate, and random-walk covariate. Across 36 model-shift combinations they achieve the best or tied-best average rank in 32 cases, cutting Absolute Error by up to 30-50% relative to Probabilistic ACC.

Overall, by marrying structural reweighting with neighbourhood cues, the paper shows that ACC can be re-tooled to deliver reliable, shift-robust quantification on real-world graphs, establishing SIS + NACC as the first practical framework for graph quantification under covariate shift.

**Questions:**

1. Could you (a) add at least one naturally shifted graph - e.g. a time-stamped social network split by month, or a sampled Web graph - and (b) report SIS + NACC performance there?
2. Is it possible to add a sensitivity study ($\lambda$, L, alternative kernels)?

**Ethical Concerns:**

["NO or VERY MINOR ethics concerns only"]

**Final Justification:**

* **Overall reaction**

  * Thanks for the thorough rebuttal; most of my methodological concerns are now well-addressed.
  * The added Twitch-Gamers experiment and kernel-sensitivity appendix materially strengthen the empirical story.
  * One issue - breadth of baselines - remains only partially resolved.

* **Positives after rebuttal**

  * *Real-world shift*: New Twitch results show SIS can help under natural covariate shift; this boosts confidence in external validity.
  * *Kernel robustness*: Section B’s $\lambda$-sweep and alternative-kernel study demonstrate that PPR is a reasonable default and that performance is not hyper-fragile.
  * *Scalability*: Complexity analysis plus the 168k-node run suggest practicality for medium-sized graphs.
  * *Clarified assumptions*: Explanation of how NACC copes with local heterophily clarifies when it helps and when it can hurt.

* **Residual concerns / suggestions**

  * *Baseline scope*: I still believe at least one distribution-matching or deep-quantification baseline would make the empirical claim more compelling. Even a compact ablation on one dataset would suffice.
  * *Appendix reliance*: Key evidence (kernel study, scalability) lives only in the supplement. A short summary in the main paper would aid readers.
  * *Binary vs. multi-class*: Twitch experiment is binary; it would be nice to confirm gains on a multi-class natural shift, though I acknowledge space limits.

* **Recommendation update**

  * I raise my overall score to **Accept**- the new evidence tips the balance in favor of the paper, despite the still-narrow baseline comparison.

**Limitations:**

Yes

**Quality:**

3

**Strengths And Weaknesses:**

## Strengths

**First to tackle covariate-shifted graph quantification:** The paper is the first to lift Adjusted Classify & Count (ACC) from the prior-probability-shift regime to full (structural) covariate shift on graphs, filling a clear gap in quantification research.

**Elegant ACC generalisation (SIS):** Structural Importance Sampling rewrites the test-set confusion matrix as a covariate-shift re-weighting and estimates the density ratio with personalised-PageRank kernels, gracefully collapsing to vanilla ACC when constant kernels are used.

**Neighbourhood cue for identifiability (NACC):** Appending the majority 1-hop label breaks class-confusion collinearity without retraining the base classifier, neatly exploiting homophily yet reverting to standard ACC in purely homophilic graphs.

**Plug-and-play, classifier-agnostic design:** SIS + NACC drop straight into the standard constrained-least-squares ACC solver and work on four very different backbones (MLP, GCN, GAT, APPNP) without any model-specific tuning.

**Consistent empirical gains across shifts:** On five benchmarks and three synthetic shift types, the new quantifiers attain the best or tied-best average rank in 32 / 36 model–shift combinations, cutting Absolute Error by up to 30–50 % over Probabilistic ACC.

**Graceful degradation & interpretability:** Because both upgrades reduce to classic ACC when their extra signal is irrelevant, they inherit ACC’s transparent confusion-matrix interpretation and do not hurt performance under pure PPS.

---

## Weaknesses

**Synthetic evaluation only:** All shifts are injected (Zipf PPS, BFS, RW) on node-classification datasets; the authors explicitly note the absence of real graph-quantification benchmarks, so external validity remains uncertain.

**Kernel sensitivity and hyper-parameter exposure:** SIS assumes a local random-walk sampling process and relies on a PPR kernel with a user-set $\lambda$ and walk length L; in BFS shifts the kernel is acknowledged to be a mismatch, hinting that performance may hinge on careful kernel engineering.

**Homophily dependency and noise trade-off:** NACC presumes that neighbours share labels; on heterophilic or sparsely connected graphs its extra confusion features may be noisy, and the authors concede that adding deeper-hop information quickly degrades estimates.

**Scalability not analysed:** Computing PPR densities for every training vertex can be $O(|V|^3)$ in the worst case; the paper gives no runtime or memory study, leaving open whether SIS is feasible for million-node graphs.

**Baseline breadth limited:** Comparisons stop at CC/ACC and ENQ; distribution-matching quantifiers and recent deep quantification methods are only mentioned as future work, so the empirical case is incomplete.

---

> ### Author Rebuttal · Authors · 2025-07-31
>
> Thank you for the the well-structured and constructive review! We will go over the mentioned weaknesses in order.
>
> ### Synthetic evaluation only (Q1)
>
> Since existing node-classification benchmark datasets generally do not provide enough information to extract "natural" structural shifts, a more realistic evaluation is indeed not trivial; for this reason we decided to induce different distribution shifts synthetically on real datasets.
> Despite these difficulties, we will now attempt to provide at least some evidence towards external validity.
>
> Social network datasets are a promising source of realistic structural covariate shifts.
> The "Twitch Gamers" dataset [1] is one such dataset; it consists of ~168k vertices and ~6.8 million edges where vertices represent Twitch accounts and edges represent followership relations.
> Each vertex is annotated with the language of the corresponding user, whether the user streams explicit content and a number of other features.
>
> To provide further insights into the effectiveness of SIS under "real" covariate shift, we have conducted additional experiments using the "Twitch Gamers" dataset.
> We induced covariate shift by using a random subset of all users as training data and then selected different random subsets of the remaining users as test data where each test sample only contains users speaking a single language (EN, DE, FR, RU).
> Sampling the test data based on language induces a natural structural covariate shift, since users tend to follow users speaking the same language.
> We use the binary "explicit content" feature of the users as the target.
> The following table shows the AE achieved by different quantifiers on different language-filtered test splits.
> All quantifiers use an APPNP node classifier as their base model.
>
> |              |        EN |        DE |        FR |        RU |
> | ------------ | --------: | --------: | --------: | --------: |
> | CC           | **.0040** |     .1187 |     .1147 |     .1771 |
> | ACC          |     .0208 |     .0108 | **.0888** |     .0887 |
> | SIS-PPR ACC  |     .0107 | **.0076** |     .0899 | **.0801** |
> | SIS-PPR NACC |     .0958 |     .3991 |     .1768 |    .2553 |
>
> Since EN is the majority language in the dataset, there is barely any distribution shift between training and the EN test set; therefore, the unadjusted CC approach performs best.
> For all other languages, quantification performance is significantly improved via ACC.
> SIS with the PPR kernel further improves performance for the DE and RU test sets; on the FR test set, standard ACC performs slightly better than SIS.
> Adding NACC increases the quantification error; class identifiability is not really an issue in binary quantification.
> Overall, this experiment indicates that SIS can also deal with real-world structural covariate shift.
>
> ### Kernel sensitivity and hyper-parameter exposure (Q2)
>
> We fully agree that the kernel selection problem deserves further attention.
> For this reason, we included an extended discussion of the influence of the vertex kernel and its hyperparameters in Section B of the supplement PDF.
>
> There, we evaluate the influence of $\lambda$ on the PPR kernel and explain why a value of $\lambda$ slightly below $1$ tends to perform best (Section B.2).
> Additionally, we explain why the PPR kernel is a good "default" choice for SIS (Section B.1).
>
> The general appropriateness of the PPR kernel is supported empirically by Table S1 of the supplement.
> There, we compare PPR against an alternative shortest-path-based kernel, which, at first glance, might appear to be a better match than PPR for the BFS-based covariate shift.
> Despite the apparent mismatch between PPR and BFS sampling, the PPR kernel outperforms the shortest-path kernel.
>
> Last, an evaluation of the influence of the hyperparameters of the shortest-path kernel is provided in Section B.3 of the supplement.
>
> ### Homophily dependency and noise trade-off
>
> While, both, SIS and NACC are designed for quantification problems on graphs that are (mostly) homophilic, no strict formal assumptions on homophily are made.
> In fact, NACC is motivated by the fact that even homophilic graphs can have non-homophilic subgraphs.
>
> In such non-homophilic subgraphs (see, for example, class/cluster 7 in Figure 1 of the main paper), columns in the confusion matrix $C$ can become collinear, turning quantification into an ill-posed problem.
> NACC addresses this issue by incorporating neighborhood information; this effectively allows NACC to distinguish between homophilic and heterophilic subgraphs.
>
> To summarize, while we assume homophily in general, NACC can help with local violations of this homophily assumption.
>
> Last, to avoid confusion, we want to clarify that the reason why second- or third-order vertex neighborhoods in NACC can degrade performance is not due to homophily or heterophily but because of the resulting sample sizes for the confusion matrix estimate.
>
> ### Scalability not analysed
>
> An analysis of the runtime of SIS and NACC was omitted from the main paper due to the page limit.
> We discuss the time complexity and scalability of our approach in Section C of the appendix.
>
> Naively, the PPR densities can be computed in $O(|V|^{3 \log_2 L})$.
> To apply SIS with the PPR kernel to large graphs, there are multiple options:
> 1. The kernel is not evaluated for all vertex pairs. Thus, computing a power of the entire adjacency matrix is not necessary. Instead, only train-test-pairs have to be computed. If either the training or the test data is small, one can compute the relevant kernel values in $O((|V|^2 \cdot \min ( |D_L|, |\mathcal{V}_U|))^{\log_2 L})$.
> 2. For densely connected graph, matrix multiplication can be sped up using Strassen's algorithm.
> 3. For sparse graphs, performance can be further improved via sparse-sparse or sparse-dense matrix multiplications. Combined with option 1, the relevant part of the PPR kernel can be computed in $O((|\mathcal{E}| \cdot \min ( |D_L|, |\mathcal{V}_U|))^{\log_2 L})$ via sparse-dense matrix multiplications.
>
> Using these optimizations, we were able to apply SIS to a graph with ~168k vertices (see rebuttal to KYhC) on a consumer PC with 64GB memory.
> Assuming sparsity, computing a kernel for a graph with over 1 million vertices is feasible.
>
> ### Baseline breadth limited
>
> We did not include quantification results for other quantification methods to allow for a fair comparison.
> To demonstrate the effectiveness of SIS and NACC, which are extensions of (P)ACC, they should be compared against their respective "base" quantification method.
>
> Nonetheless, we agree that an extension of our work to other quantification methods, like distribution matching, would indeed be interesting.
> In fact, since the submission, we have finished a follow-up work which generalizes SIS to the distribution matching setting.
> Overall, we found, that SIS also performs strongly in the distribution-matching setting.
>
> We hope that we were able to answer your questions and address your suggestions. Thank you for your time and effort!
>
> ---
>
> 1. Rozemberczki, B., Sarkar, R.: Twitch Gamers: a Dataset for Evaluating Proximity Preserving and Structural Role-based Node Embeddings (2021).

---

> > ### Comment · Area_Chair_5Bqz · 2025-08-04
> >
> > Dear reviewer,
> >
> > Please engage in the discussion with the authors. The discussion period will end in a few days.

---

> > ### Comment · Reviewer_GwGf · 2025-08-05
> > **Rebuttal Feedback #1**
> >
> > * **Overall reaction**
> >
> >   * Thanks for the thorough rebuttal; most of my methodological concerns are now well-addressed.
> >   * The added Twitch-Gamers experiment and kernel-sensitivity appendix materially strengthen the empirical story.
> >   * One issue - breadth of baselines - remains only partially resolved.
> >
> > * **Positives after rebuttal**
> >
> >   * *Real-world shift*: New Twitch results show SIS can help under natural covariate shift; this boosts confidence in external validity.
> >   * *Kernel robustness*: Section B’s $\lambda$-sweep and alternative-kernel study demonstrate that PPR is a reasonable default and that performance is not hyper-fragile.
> >   * *Scalability*: Complexity analysis plus the 168k-node run suggest practicality for medium-sized graphs.
> >   * *Clarified assumptions*: Explanation of how NACC copes with local heterophily clarifies when it helps and when it can hurt.
> >
> > * **Residual concerns / suggestions**
> >
> >   * *Baseline scope*: I still believe at least one distribution-matching or deep-quantification baseline would make the empirical claim more compelling. Even a compact ablation on one dataset would suffice.
> >   * *Appendix reliance*: Key evidence (kernel study, scalability) lives only in the supplement. A short summary in the main paper would aid readers.
> >   * *Binary vs. multi-class*: Twitch experiment is binary; it would be nice to confirm gains on a multi-class natural shift, though I acknowledge space limits.
> >
> > * **Recommendation update**
> >
> >   * I raise my overall score to **Accept**- the new evidence tips the balance in favor of the paper, despite the still-narrow baseline comparison.

---

### Official Review · Reviewer_KYhC · 2025-07-03

**Clarity:** 4
**Significance:** 2
**Originality:** 2
**Rating:** 4
**Confidence:** 3

**Summary:**

The paper develops a method to estimate the label distribution in a node classification graph problem, under covariate shift.
Two adjustments that are tailored to graph data are given. One applies a density ratio correction based on a kernel density estimator, where the kernel is derived from the graph. The second applies an adjustment to label distributions based on homophily between neighboring vertices, in order to alleviate collinearity of label distributions. These adjustments are applied on top of standard adjustments applied for label shifts.

Experiments on several node classification datasets using various classifiers are given, under label shift, and two types of covariate shifts induced by sampling using random walks on the graph. The methods show an overall improved performance over the baseline estimation method.

**Questions:**

I've listed my suggestions in the strengths and weaknesses part. Otherwise, the paper is well written and I appreciate presentation.
More compelling datasets and baselines could make me consider raising my score, though it will still be a borderline case in my opinion.

**Ethical Concerns:**

["NO or VERY MINOR ethics concerns only"]

**Final Justification:**

Following the author response, which includes additional experiments to address concerns raised in the reviews, I've decided to raise my score. My confidence in this score is not high, as I am not an expert on the topic, but I believe the score is justified.

**Limitations:**

Yes

**Paper Formatting Concerns:**

No concerns

**Quality:**

3

**Strengths And Weaknesses:**

Strengths:

The paper is nicely written, self contained and the methods are principled, simple, and seem effective. Experiments are compelling as they cover a nice variety of settings.

Weaknesses:

Applying a density ratio correction to adjust for covariate shift is a rather basic technique and even though the paper gives a nice presentation and a solid solution, I am not sure its novelty merits a publication at NeurIPS.
At the very least, I would expect a baseline that applies a density ratio correction not based on random walks, e.g. one that learns a probabilistic model to distinguish between the features of the source and target dataset. Even if the suggested method outperforms such a baseline, it wouldn't be surprising since the covariate shift induced in the experiments is tailored to the kernel in proposed method. A much more compelling experiment would consider a covariate shift that occurs naturally in graph data, e.g. by considering two different datasets. It is understandable that it may be difficult to find such data, but there should be a through discussion about it with experiments that examine baselines that handle covariate shift and datasets with shifts that do not adhere to the assumptions underlying the kernel.

---

> ### Author Rebuttal · Authors · 2025-07-31
>
> We thank the reviewer for their constructive feedback!
>
> ### Inclusion of a feature-based baseline
>
> > I would expect a baseline that applies a density ratio correction not based on random walks, e.g. one that learns a probabilistic model to distinguish between the features of the source and target dataset.
>
> Thank you for the suggestion! The goal of SIS is to address *structural covariate shift*; a density ratio correction based on vertex features therefore would not be well-motivated from a theoretical perspective.
>
> Nonetheless, we agree, that a feature-based importance sampling approach is an interesting additional baseline.
> To this end, we have conducted additional experiments with a simple feature-based reweighting approach using the cosine similarity between vertex feature vectors as the kernel $k$.
>
> The following table compares the AE of this feature-based quantifier (*SIS-Feat*) with other quantifiers using an APPNP model (best results highlighted in **bold**):
>
> | Model & Shift | Quantifier      | CoraML    | CiteSeer  | A. Photos | A. Computers |
> | ------------- | --------------- | --------- | --------- | --------- | ------------ |
> | APPNP PPS     | PCC             | .0374     | .0214     | .0318     | .0390        |
> |               | PACC            | .0217     | .0184     | **.0124** | .0256        |
> |               | SIS-PPR PACC    | **.0203** | **.0171** | .0130     | **.0249**    |
> |               | *SIS-Feat PACC* | .0212     | .0183     | **.0124** | .0255        |
> | APPNP BFS     | PCC             | .0374     | .0737     | .0271     | .0468        |
> |               | PACC            | .0217     | .0603     | .0225     | .0430        |
> |               | SIS-PPR PACC    | **.0203** | **.0574** | **.0213** | **.0395**    |
> |               | *SIS-Feat PACC* | 0.212     | .0985     | **.0220** | .0425        |
> | APPNP RW      | PCC             | .0465     | .0659     | .0293     | .0504        |
> |               | PACC            | .0527     | .0541     | .0282     | .0452        |
> |               | SIS-PPR PACC    | **.0448** | **.0501** | **.0255** | **.0418**    |
> |               | *SIS-Feat PACC* | .0515     | .0744     | .0279     | .0452        |
>
> The structure-unaware *SIS-Feat* baseline (generally) performs worse than the structure-aware SIS-PPR.
> Unsurprisingly, this confirms that structure-based vertex kernels are better suited than feature-based kernels to account for (structural) covariate shifts.
>
> ### Evaluation of "real" covariate shift
>
> > A much more compelling experiment would consider a covariate shift that occurs naturally in graph data, e.g. by considering two different datasets. It is understandable that it may be difficult to find such data [...]
>
> Since existing node-classification benchmark datasets generally do not provide enough information to extract "natural" structural shifts, a more realistic evaluation is indeed not trivial; in our experiments, we therefore induced different distribution shifts synthetically on real datasets.
>
> Using two (disjoint) training and test datasets as a source of natural covariate shifts is, unfortunately, not feasible because SIS relies on kernel density estimates across the training and test.
> If the training and test vertices are not connected, no meaningful structure-based vertex kernel can be defined.
>
> Following your suggestion, we thought about other potential sources of realistic structural covariate shifts and found social network datasets to be promising candidates.
> The "Twitch Gamers" dataset [1] is one such dataset; it consists of ~168k vertices and ~6.8 million edges where vertices represent Twitch accounts and edges represent followership relations.
> Each vertex is annotated with the language of the corresponding user, whether the user streams explicit content and a number of other features.
>
> To provide further insights into the effectiveness of SIS under "real" covariate shift, we have now conducted additional experiments using the "Twitch Gamers" dataset.
> We induced covariate shift by using a random subset of all users as training data and then selected different random subsets of the remaining users as test data where each test sample only contains users speaking a single language (EN, DE, FR, RU).
> Sampling the test data based on language induces a natural structural covariate shift, since users tend to follow users speaking the same language.
> We use the binary "explicit content" feature of the users as the target.
> The following table shows the AE achieved by different quantifiers on different language-filtered test splits.
> All quantifiers use an APPNP node classifier as their base model.
>
> |              |        EN |        DE |        FR |        RU |
> | ------------ | --------: | --------: | --------: | --------: |
> | CC           | **.0040** |     .1187 |     .1147 |     .1771 |
> | ACC          |     .0208 |     .0108 | **.0888** |     .0887 |
> | SIS-PPR ACC  |     .0107 | **.0076** |     .0899 | **.0801** |
> | SIS-PPR NACC |     .0958 |     .3991 |     .1768 |    0.2553 |
>
> Since EN is the majority language in the dataset, there is barely any distribution shift between training and the EN test set; therefore, the unadjusted CC approach performs best.
> For all other languages, quantification performance is significantly improved via ACC.
> SIS with the PPR kernel further improves performance for the DE and RU test sets; on the FR test set, standard ACC performs slightly better than SIS.
> Adding NACC increases the quantification error; class identifiability is not really an issue in binary quantification.
> Overall, this experiment indicates that SIS can also deal with real-world structural covariate shift.
>
> ### On kernel selection
>
> > there should be a through discussion about it [datasets and kernels?] with experiments that examine baselines that handle covariate shift and datasets with shifts that do not adhere to the assumptions underlying the kernel
>
> A discussion of kernel selection and its influence on the effectiveness of SIS can be found in Section B of the supplementary PDF.
>
> The effectiveness of SIS does not strictly depend on the accuracy of the density estimate $\hat{q}$; more precisely, the confusion matrix estimate $\hat{C}$ can be accurate, even if $\hat{q}$ deviates from $q$.
> Ultimately, we are only interested in whether the predictive quality of the classifier $h$ on $\hat{q}$ is similar to the predictive quality of $h$ on $\hat{q}$.
> From this perspective, we argue that the PPR kernel can be seen as a good "default" choice because it (approximately) describes the GNN prediction similarity between vertices.
>
> Further details on this can be found in the supplement (see Table S1).
> We compared PPR against an alternative shortest-path-based kernel, which, at first glance, might appear to be a better match than PPR for the BFS-based covariate shift.
> Despite the apparent mismatch between PPR and BFS sampling, the PPR kernel outperforms the shortest-path kernel.
>
> ### Novelty
>
> >  even though the paper gives a nice presentation and a solid solution, I am not sure its novelty merits a publication at NeurIPS
>
> We appreciate the feedback! While the general idea of importance sampling is, of course, not novel by itself, its application in the context of quantification learning is.
>
> To our knowledge, we are the first to systematically solve the quantification problem in the presence of covariate shift; prior work in the field of quantification learning is mostly concerned with prior probability shift (PPS).
> Moreover, the ideas described in this paper could, in principle, be extended to address covariate shift in quantification problems outside of the graph domain by defining an appropriate kernel for density estimation.
> Overall, we thus believe that our contribution is of relevance not only for graph quantification learning but for the field of quantification learning as a whole.
>
> We hope that we were able to answer your questions. Thank you, once again, for your comments!
>
> ---
>
> 1. Rozemberczki, B., Sarkar, R.: Twitch Gamers: a Dataset for Evaluating Proximity Preserving and Structural Role-based Node Embeddings (2021).

---

> > ### Comment · Area_Chair_5Bqz · 2025-08-04
> >
> > Dear reviewer,
> >
> > Please engage in the discussion with the authors. The discussion period will end in a few days.

---

> > ### Comment · Reviewer_KYhC · 2025-08-04
> > **Post Rebuttal Update**
> >
> > Thank you for the detailed response and additional experiments.
> >
> > They help answer some of my questions and I will raise my score accordingly. I will also emphasize that my confidence in this recommendation is somewhat low since I am not an expert on this specific problem.

---

> > > ### Author Response · Authors · 2025-08-08
> > >
> > > Thank you! We are glad you found our response helpful.

---

### Note · Authors · 2025-08-15

## Summary of Strengths
1. *Novelty:* Reviewers GwGf and ZwZ8 acknowledge that our work is the first to tackle the covariate-shift problem for graph quantification.
2. *Method:* Reviewer KYhC found our proposed SIS and NACC methods to be "principled, simple, and seem[ingly] effective". SIS is described as an "Elegant ACC generalisation" (GwGf) with a "Plug-and-play, classifier-agnostic design" (GwGf).
3. *Experiments:* The reviewers highlight that our experiments are "thorough" (ZwZ8) and "compelling as they cover a nice variety of settings" (KYhC), with "Consistent empirical gains across shifts" (GwGf).
4. *Presentation:* Reviewer KYhC found our paper to be "nicely written", reviewer ohSm acknowledges a "Clear problem demonstration" with "Good background writing".

## Summary of Suggestions
1. We have shown the effectiveness of SIS and NACC on synthetically induced shifts. All reviewers suggest that the evaluation could be further strengthened by also considering natural shifts. We took up this suggestion and conducted additional experiments on the "Twitch Gamers" dataset, on which our approach also performed well. Reviewer GwGf acknowledged that "this boosts confidence in external validity" and "materially strengthens the empirical story".
2. Reviewers GwGf and ZwZ8 asked for additional information on kernel selection. To some extent, Section B of the supplement already answers this question, explaining how kernels affect quantification and providing experimental results on the influence of kernel hyperparams. Furthermore, in our rebuttal to reviewer ZwZ8, we provided guidance for how the SIS kernel should be chosen. Reviewer GwGf found "that performance [of SIS] is not hyper-fragile" wrt. kernel choice.
3. Reviewers GwGf and ZwZ8 raised concerns regarding scalability. Overall, we found SIS and NACC to be applicable even for large graphs, as demonstrated by our experiment on the "Twitch Gamers" dataset with ~168k vertices. A formal complexity analysis can be found in Section C of the supplement.

## Planned Changes
1. We plan to include results for natural covariate shifts (Twitch Gamers).
2. Following Rebuttal Feedback #1 by GwGf, we plan to add a summary of the results from the supplement to the main paper.
3. Following a suggestion by reviewer KYhC, we implemented and evaluated a new feature-based quantification baseline. We plan to include the results for this new baseline in our evaluation section.

We want to thank all reviewers for their feedback!

---

### Decision · Program_Chairs · 2025-09-17

**Decision:**

Accept (poster)

**Comment:**

This paper studies quantification learning (i.e., predicting the label distribution of a set of instances) for graphs (where instances are the graph nodes). This paper extends the adjusted classify and count (ACC) method for graph inputs with structural importance sampling. Notably, the method is effective under structural covariate shift. The second contribution is neighborhood-aware ACC, which improves quantification for non-homophilic graphs.

The paper received mixed reviews. However, those leaning towards rejection are based on a misunderstanding of the paper context and purpose. The remaining concerns were successfully addressed by the authors. Overall, I find that this paper tackles an under-researched but important problem (quantification on graphs) with a sufficiently innovative methodology. Therefore, I recommend acceptance, provided that all the reasonable suggestions made by the reviewers are implemented in the revised version.